# Product distribution, kinetics, and aerosol formation from the OH oxidation of dimethyl sulfide under different RO$_2$ regimes

**Qing Ye**[1,a]**, Matthew B. Goss**[1]**, Jordan E. Krechmer**[2]**, Francesca Majluf**[2]**, Alexander Zaytsev**[3]**, Yaowei Li**[3]**, Joseph R. Roscioli**[4]**, Manjula Canagaratna**[2]**, Frank N. Keutsch**[3,5,6]**, Colette L. Heald**[1]**, and Jesse H. Kroll**[1]

[1]Department of Civil and Environmental Engineering, Massachusetts Institute of Technology, Cambridge, Massachusetts 02139, United States
[2]Center for Aerosol and Cloud Chemistry, Aerodyne Research Incorporated, Billerica, Massachusetts 01821, United States
[3]John A. Paulson School of Engineering and Applied Sciences, Harvard University, Cambridge, Massachusetts 02138, United States
[4]Center for Atmospheric and Environmental Chemistry, Aerodyne Research Incorporated, Billerica, Massachusetts 01821, United States
[5]Department of Chemistry and Chemical Biology, Harvard University, Cambridge, Massachusetts 02138, United States
[6]Department of Earth and Planetary Sciences, Harvard University, Cambridge, Massachusetts 02138, United States
[a]now at: Atmospheric Chemistry Observations and Modeling, National Center for Atmospheric Research, Boulder, Colorado 80301, United States

**Correspondence:** Qing Ye (qingye@ucar.edu), Jesse H. Kroll (jhkroll@mit.edu)

**Abstract.** The atmospheric oxidation of dimethyl sulfide (DMS) represents a major natural source of atmospheric sulfate aerosols. However, there remain large uncertainties in our understanding of the underlying chemistry that governs the product distribution and sulfate yield from DMS oxidation. Here, chamber experiments were conducted to simulate gas-phase OH-initiated oxidation of DMS under a range of reaction conditions. Most importantly, the bimolecular lifetime ($\tau_{bi}$) of the peroxy radical CH$_3$SCH$_2$OO was varied over several orders of magnitude, enabling the examination of the role of peroxy radical isomerization reactions on product formation. An array of analytical instruments was used to measure nearly all sulfur-containing species in the reaction mixture, and results were compared with a near-explicit chemical mechanism. When relative humidity was low, "sulfur closure" was achieved under both high-NO ($\tau_{bi} < 0.1$ s) and low-NO ($\tau_{bi} > 10$ s) conditions, though product distributions were substantially different in the two cases. Under high-NO conditions, approximately half the product sulfur was in the particle phase, as methane sulfonic acid (MSA) and sulfate, with most of the remainder as SO$_2$ (which in the atmosphere would eventually oxidize to sulfate or be lost to deposition). Under low-NO conditions, hydroperoxymethyl thioformate (HPMTF, HOOCH$_2$SCHO), formed from CH$_3$SCH$_2$OO isomerization, dominates the sulfur budget over the course of the experiment, suppressing or delaying the formation of SO$_2$ and particulate matter. The isomerization rate constant of CH$_3$SCH$_2$OO at 295 K is found to be 0.13±0.03 s$^{-1}$, in broad agreement with other recent laboratory measurements. The rate constants for the OH oxidation of key first-generation oxidation products (HPMTF and methyl thioformate, MTF) were also determined ($k_{OH+HPMTF} = 2.1 \times 10^{-11}$ cm$^3$ molec.$^{-1}$ s$^{-1}$, $k_{OH+MTF} = 1.35 \times 10^{-11}$ cm$^3$ molec.$^{-1}$ s$^{-1}$). Product

measurements agree reasonably well with mechanistic predictions in terms of total sulfur distribution and concentrations of most individual species, though the mechanism overpredicts sulfate and underpredicts MSA under high-NO conditions. Lastly, results from high-relative-humidity CE1 conditions suggest efficient heterogenous loss of at least some gas-phase products.

## 1   Introduction

Dimethyl sulfide (DMS), emitted by marine phytoplankton, is an important natural source of sulfur to the atmosphere (Kloster et al., 2006; Lana et al., 2011). The atmospheric oxidation of DMS represents a dominant source of non-sea-salt sulfate aerosols and as such can play an important role in global aerosol climate effects (Charlson et al., 1987; Rap et al., 2013). The chemistry by which DMS oxidizes to form sulfate is highly complex: the mechanism includes multiple branch points and intermediate species, and many reaction rates and product yields are uncertain and/or highly dependent on reaction conditions (Barnes et al., 2006; Hoffmann et al., 2016). As a result, many large-scale models adopt a highly simplified DMS chemistry with fixed $SO_2$ yields, usually without inclusion of other intermediates (Chin et al., 1996; Huijnen et al., 2010; Kloster et al., 2006; Lamarque et al., 2012). Such a simplified approach may lead to errors in predicted aerosol radiative effects, in the past, present, and future atmospheres (Fung et al., 2022).

The major daytime sink of DMS is its reaction with OH radicals. The detailed DMS + OH reaction scheme is shown in Fig. 1. A key branch point in DMS + OH is the methylthiomethylperoxy radical ($CH_3SCH_2OO$) formed from H-atom abstraction followed by $O_2$ addition. The subsequent chemistry of this radical plays a determining role in the overall product distribution and thus likely influences the amount of sulfate aerosols that is ultimately formed. As with all large $RO_2$ species, $CH_3SCH_2OO$ radicals may undergo bimolecular reactions (with NO and $HO_2$) or unimolecular reaction via a recently identified (Berndt et al., 2019; Veres et al., 2020; Wu et al., 2015; Ye et al., 2021; Jernigan et al., 2022a) isomerization channel:

$$CH_3SCH_2OO + NO \rightarrow CH_3SCH_2O + NO_2 \qquad (R1)$$
$$CH_3SCH_2OO + HO_2 \rightarrow CH_3SCH_2OOH + O_2 \qquad (R2)$$
$$CH_3SCH_2OO \rightarrow CH_2SCH_2OOH. \qquad (R3)$$

The $CH_3SCH_2O$ radical formed from the NO pathway (Reaction R1) forms $SO_2$, sulfate, and methane sulfonic acid (MSA) (Barnes et al., 2006). The alkyl radical derived from Reaction (R3) will react with $O_2$ to form $OOCH_2SCH_2OOH$, which will undergo a second isomerization reaction at a rate substantially faster than that of Reaction (R3) (Wu et al., 2015; Crounse et al., 2013), forming hydroperoxymethyl thioformate (HPMTF, $HOOCH_2SCHO$), as shown in Fig. 1. In addition to Reactions (R1–R3), $CH_3SCH_2OO$ may also react with other $RO_2$ radicals (Barnes et al., 2006), though this process is likely to be minor under atmospheric conditions.

The branching fraction of the $CH_3SCH_2OO$ radical depends on the concentrations of NO and $HO_2$ and the rate constants of Reactions (R1–R3). The rate constant for the isomerization reaction, $k_{isom}$, is particularly uncertain, as values determined in previous studies span a very wide range, from $\sim 0.04$ to $\sim 2\,\mathrm{s}^{-1}$ near room temperature (Berndt et al., 2019; Veres et al., 2020; Wu et al., 2015; Ye et al., 2021; Jernigan et al., 2022a). This highlights a major challenge in predicting $CH_3SCH_2OO$ branching and the subsequent aerosol formation, both in the pristine atmosphere and in environments affected by anthropogenic emissions.

Most previous experimental studies investigating DMS oxidation have examined individual products and reaction steps in isolation (Barnes et al., 2006; Berndt et al., 2019; Jernigan et al., 2022a; Mihalopoulos et al., 1992; Patroescu et al., 1996); very few studies of the entire multiphase and multistep reaction system have been conducted, especially under conditions in which the recently discovered isomerization pathway (Reaction R3) may compete. Therefore, there have been relatively few experimental tests of our overall understanding of the reaction system, by comparison against predictions by state-of-the-art reaction mechanisms. Recently, we conducted laboratory measurements of a broad suite of organic sulfur products and sulfate aerosols from DMS + OH and estimated $k_{isom}$ to be $0.09\,\mathrm{s}^{-1}$ ($0.03–0.3\,\mathrm{s}^{-1}$, $1\sigma_g$) (Ye et al., 2021); however this was for a single reaction condition only ($< 5\,\%$ relative humidity, $\sim 1$ ppb NO), and $SO_2$ (a major inorganic sulfur-containing product) was not measured.

Here we extend our previous work by conducting a series of chamber experiments of DMS + OH under a wide range of values of the $CH_3SCH_2OO$ bimolecular lifetime ($\tau_{bi}$) and comprehensively characterizing sulfur-containing products (organic and inorganic, gas-phase and particulate), with the aim of accounting for all (or nearly all) reacted sulfur. Such "sulfur closure" measurements enable direct comparisons with predictions from a mechanistic model, in order to assess our current mechanistic understanding and identify possible gaps in this understanding. These measurements also enable the determination of key kinetic parameters in the reaction systems. In one experiment, we vary $\tau_{bi}$ over a wide range to estimate the $k_{isom}$ of the $CH_3SCH_2OO$ radical, obtaining a $k_{isom}$ with a much smaller uncertainty range than in our previous study. The rate constants for the OH oxidation of key first-generation oxidation products (HPMTF and methyl thioformate, MTF) are also determined. Lastly,

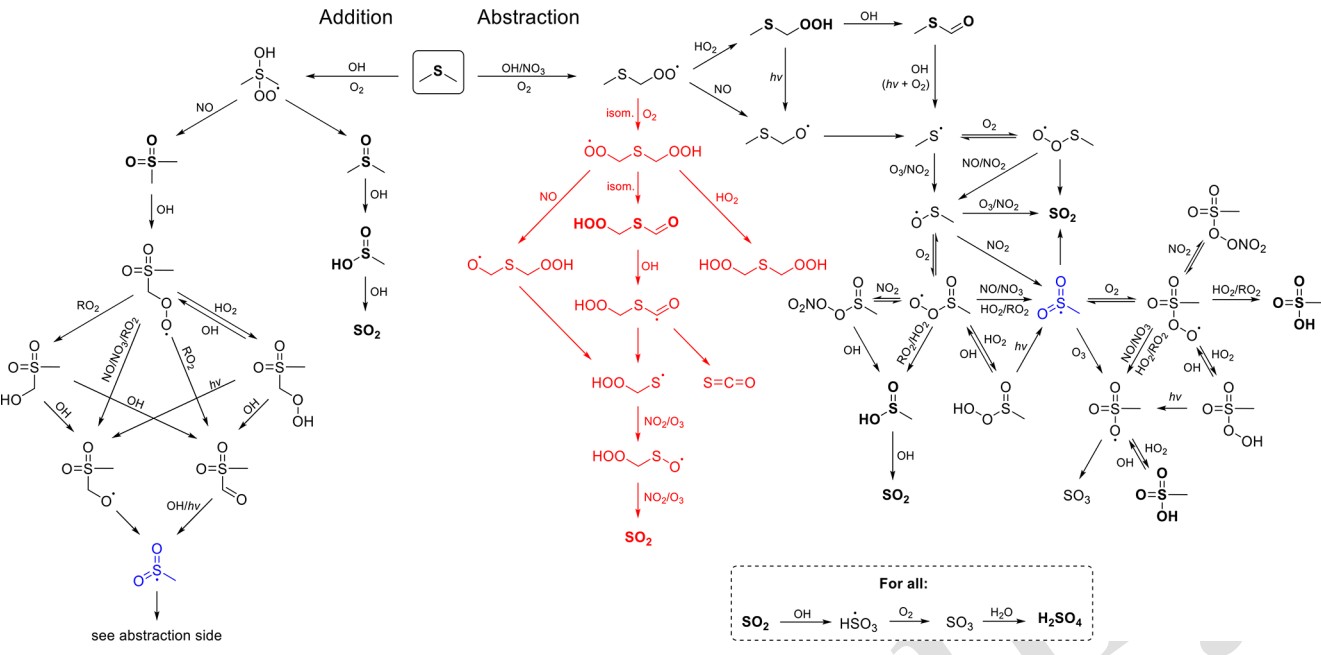

**Figure 1.** Gas-phase DMS + OH oxidation mechanism. Measured closed-shell products are shown in bold. Reactions in black are taken from MCM (Master Chemical Mechanism); reactions in red, related to hydroperoxymethyl thioformate (HPMTF, HOOCH$_2$SCHO) chemistry, are taken from Wu et al. (2015). Products that do not contain sulfur are not shown. The CH$_3$SO$_2$ radical (marked in blue) represents a link between addition and abstraction pathway products. Note that several products are shown multiple times.

we investigate the effect of relative humidity (RH) on the DMS + OH product distributions.

## 2   Method and materials

Experiments were conducted in a 7.5 m$^3$ temperature-controlled environmental chamber, held at 295 K (Hunter et al., 2014). The chamber is surrounded by 48 ultraviolet lights (Q-Lab) with a peak irradiance at 340 nm. Before each experiment, the chamber was flushed by zero air (AADCO, 737 series) for at least 12 h to ensure a clean gas and particle background. Throughout the course of each experiment, a constant flow of zero air was introduced into the chamber to replenish the flow drawn by the instruments. For high-RH experiments, the replenishment flow was first sent through a bubbler filled with Milli-Q water before entering the chamber. The rate of chamber dilution was derived by measuring the decay of acetonitrile, injected at low concentrations (5 ppb) in the beginning of each experiment. The overall dilution lifetime was approximately 10 h. Concentrations of all species reported below have been corrected for dilution.

The evolving chemical composition of the reaction mixture was monitored by a suite of real-time instruments located outside the chamber. The Supplement provides instrument details, as well as the sulfur species detected by each instrument (Table S2). Briefly, DMS and lightly oxygenated gaseous species were measured by a Vocus proton-transfer-reaction time-of-flight mass spectrometer (Vocus PTR-MS,

Aerodyne Research Inc.) (Krechmer et al., 2018). More oxygenated gaseous species were measured by an iodide time-of-flight chemical ionization mass spectrometer (I$^-$ CIMS, Aerodyne Research Inc.) (Lee et al., 2014) and an ammonium time-of-flight chemical ionization mass spectrometer (NH$_4^+$ CIMS, Ionicon Analytik) (Zaytsev et al., 2019). SO$_2$ was detected by a compact tunable infrared laser direct absorption spectrometer (TILDAS, Aerodyne Research Inc.) (McManus et al., 2011, 1995). Particle-phase products, namely sulfate and MSA, were measured by an aerosol mass spectrometer (AMS, Aerodyne Research Inc.) (DeCarlo et al., 2006). The quantification of MSA was determined from the AMS tracer ion CH$_3$SO$_2^+$ (see the Supplement); this ion is believed to be unique to MSA (or methyl sulfonate), with negligible contributions from other sulfur-containing species (Hodshire et al., 2019; Huang et al., 2015). Our multi-instrument approach enables the measurement of essentially all closed-shelled sulfur products known in the DMS oxidation mechanism, except for carbonyl sulfide (OCS), which accounts for a very small (less than a couple percent) sulfur yield from DMS oxidation (Barnes et al., 1994; Jernigan et al., 2022a). Complementary instruments include an ozone monitor (2B Technologies), a NO–NO$_2$–NO$_x$ analyzer (Thermo Scientific), a scanning mobility particle sizer (TSI), and a temperature and RH sensor (TE Connectivity). More details of the instruments, including their calibrations and measurement uncertainties, are provided in the Supplement.

**Table 1.** Summary of experimental conditions.

| Exp. no. | Precursor(s)[a] | OH precursor | $[OH]_{avg}$ (molec. cm$^{-3}$) | Dominant $RO_2$ fate | $\tau_{bi}$ (s)[b] | Seed particles | RH | Corresponding figure(s) |
|---|---|---|---|---|---|---|---|---|
| 1 | $\sim 70$ ppb DMS-$^{12}C_2$ | HONO | $\sim 1 \times 10^7$ | $RO_2 + NO$ | <0.1 | $NH_4NO_3$ | dry, <5 % | Figs. 2a, S3 |
| 2a[c] | $\sim 40$ ppb DMS-$^{12}C_2$, $\sim 40$ ppb DMS-$^{13}C_2$ | $H_2O_2$ | $\sim 1.5 \times 10^6$ | $RO_2$ isom. | >10 | $NH_4NO_3$ | dry, <5 % | Figs. 2b, S4 |
| 2b[c] | | NO and $H_2O_2$ | $\sim 4 \times 10^6$ | $RO_2 + NO$ | <0.1 | $NH_4NO_3$ | dry, <5 % | Figs. 4, S9 |
| 3[d] | $\sim 35$ ppb DMS-$^{12}C_2$, $\sim 35$ ppb DMS-$^{13}C_2$ | $H_2O_2$ and HONO | $\sim 5 \times 10^6$ | $RO_2$ isom. $RO_2 + NO$ | < 0.1–10 | $NH_4NO_3$ | dry, <5 % | Figs. 3a, S6, S8 |
| 4 | $\sim 70$ ppb DMS-$^{12}C_2$ | HONO | $\sim 1 \times 10^7$ | $RO_2 + NO$ | <0.1 | NaCl[e] | 65 ± 3 % | Figs. 5, S11a |
| 5 | $\sim 40$ ppb DMS-$^{12}C_2$, $\sim 40$ ppb DMS-$^{13}C_2$ | $H_2O_2$ and HONO | $\sim 6 \times 10^6$ | $RO_2$ isom. | >1 | $NaNO_3$ | 65 ± 3 % | Figs. 5, S11b |

[a] To better separate HPMTF from $N_2O_5$, DMS-$^{13}C_2$ was used in low-NO experiments. [b] Bimolecular lifetime of the $CH_3SCH_2OO$ radical, calculated as $\tau_{bi} = (k_{RO_2+HO_2}[HO_2] + k_{RO_2+NO}[NO])^{-1}$. [c] Experiments 2a and b were carried out as part of a single oxidation experiment; initially (Exp. 2a) OH was generated from $H_2O_2$ photolysis (low-NO conditions), and then (Exp. 2b) 70 ppb of NO was injected into the chamber. [d] $^{13}C$ data in Experiment 3 were used to calculate $k_{isom}$; HONO was added multiple times in the experiment. [e] The vaporizer in the AMS was operated at 800 °C. AMS calibration was done separately for 800 °C.

The experiments carried out in this study are listed in Table 1. At the beginning of each experiment, DMS, the acetonitrile dilution tracer, seed particles, and the OH precursor were added to the chamber and allowed to become well mixed. Total concentrations of DMS introduced to the chamber were similar among all experiments. In dry experiments, seed particles (ammonium nitrate) were added into the chamber via first atomization followed by drying, providing surface area for condensing vapors. In high-RH experiments, seed particles (sodium chloride and sodium nitrate) were introduced without drying, remaining as liquid particles under the chamber RH. Particle condensation timescales (seconds to tens of seconds) were much shorter than the condensation timescale of low-volatility species onto the chamber wall ($\sim 2000$ s, as determined previously for this chamber, Zaytsev et al., 2019). In these experiments, non-sulfate seeds were used to avoid interferences when quantifying secondary sulfate in the aerosols. For low-RH experiments (Exp. 1–3), ammonium nitrate seed particles were used, since dry ammonium nitrate particles are expected to be chemically inert. For the high-RH high-NO experiment (Exp. 4), NaCl particles were used. As discussed below, major products are similar to those in the high-NO dry experiment, suggesting that the NaCl seed particles in Exp. 4 have little to no effect on

the product distribution in these experiments. More studies are needed to constrain the effects of different seed particles on the reactive uptake of DMS oxidation products (Jernigan et al., 2022b).

DMS was introduced by gently heating a known volume (1–2 µL) from a needle syringe, and the vapor was carried into the chamber by the dilution flow. For the long $\tau_{bi}$ experiments, in which HPMTF formation was expected (see Table 1), DMS-$^{13}C_2$ (99 atom % $^{13}C$, Millipore-Sigma) was added as the precursor in addition to unlabeled DMS (>99 %, MilliporeSigma), in order to easily distinguish HPMTF ($C_2H_4SO_3 \cdot I^-$, $m/z$ 234.893) from $N_2O_5$ ($N_2O_5 \cdot I^-$, $m/z$ 234.886) in the I$^-$ CIMS. The use of DMS-$^{13}C_2$ is expected to have little effect on the observed reaction kinetics in this study. For the high-NO (short $\tau_{bi}$) experiments, HONO (tens of parts per billion) was added as the OH precursor, by passing air over a mixture of sodium nitrite and sulfuric acid into the chamber. For low-NO (long $\tau_{bi}$) experiments, parts per million levels of $H_2O_2$ were added as the OH precursor, by vaporizing a known amount of 30 % $H_2O_2$ solution injected by a micro-syringe. The $H_2O_2$ concentration was derived based on the known photon flux in the chamber and the observed decay rate of NO. In some experiments (Exp. 2b, 3, and 5), aliquots of HONO or NO

were added in the middle of the experiment to change reaction conditions. After all reagents were well mixed ($>5$ min), the UV lights were turned on to photolyze HONO and/or $H_2O_2$, generating OH radicals and initiating the reaction. The OH concentration was estimated from the decay of DMS (using $k_{OH+DMS} = 6.97 \times 10^{-12}\,cm^3\,molec.^{-1}\,s^{-1}$) (Jenkin et al., 1997; Saunders et al., 2003) and was used to determine the equivalent atmospheric OH exposure time, assuming $[OH]_{atm} = 1.5 \times 10^6\,molec.\,cm^{-3}$.

A 0-D model (the Framework for 0-D Atmospheric Modeling, F0AM) (Wolfe et al., 2016) coupled with the Master Chemical Mechanism (MCMv3.3.1) (Jenkin et al., 1997; Saunders et al., 2003) was used to simulate gas-phase DMS oxidation in each experiment. Here, the DMS scheme in the MCM was updated primarily based on Wu et al. (2015) with the isomerization rate constant of the $CH_3SCH_2OO$ radical as $0.09\,s^{-1}$, taken from our previous work (Ye et al., 2021). The complete reaction scheme is shown in Fig. 1. Newly added reactions with rate constants beyond the original MCM scheme are listed in Table S1. Model inputs, including concentrations of the precursor, oxidant, and chamber conditions including temperature, light intensity, and dilution rate were taken directly from the measurements. The uptake or heterogeneous reactions of water-soluble species (e.g., dimethyl sulfoxide (DMSO), dimethyl sulfone ($DMSO_2$), methane sulfinic acid (MSIA), and HPMTF CE2) are not considered in this modeling, though as described below such processes may occur. In the high-NO experiments, model NO concentrations were constrained to values measured by the $NO–NO_2–NO_x$ analyzer. In the low-NO experiment (Exp. 2a) in which the sub-ppb-level NO concentration was near or below the detection limit (0.4 ppb) of the $NO_x$ analyzer, the model was used to constrain background NO concentration by matching the modeled DMS decay to the measured decay (Ye et al., 2021). The estimated [NO] in Exp. 2a was $\sim 10$ ppt.

## 3 Results and discussions

### 3.1 Comprehensive measurements of S-containing products

Figure 2a and b shows the measured product evolution from Experiments 1 and 2a under dry conditions. A range of sulfur-containing products were measured in both the gas and aerosol phases, shown as stacked colored traces. Changes in concentrations are given in parts per billion of sulfur ($\Delta$ ppb S) and are presented as a function of atmosphere-equivalent OH exposure time. Shown in grey is the amount of DMS oxidized over the course of the experiment. By the end of the experiment, only a fraction of the DMS had been consumed, since OH exposures were not high enough to fully deplete the DMS. In Exp. 1 (high-NO conditions, Fig. 2a), HONO was used as the OH precursor, and the NO was kept at $\sim 50$ ppb by continuous addition, ensuring that the dominant

fate of the $RO_2$ radicals was reaction with NO ($\tau_{bi} < 0.1$ s). After $\sim 12$ h of atmosphere-equivalent OH exposure, 104 % (100 %–124 %, $1\sigma$) of the reacted sulfur was measured as products, indicating excellent sulfur closure. The uncertainty in sulfur closure includes uncertainty in both gas-phase and particle-phase measurements (see Supplement for more details). The initial dip in the first 2 h may be due to loss of products to surfaces such as the chamber wall or sampling lines. It is likely that there is an equilibrium between the sampling line and the gas phase. This drop, of 1–2 ppb S, represents a relatively small portion of the total sulfur reacted by the end of the experiment.

Major sulfur-containing products in Exp. 1 were $SO_2$, particulate MSA, and particulate sulfate, with 48 % of the product sulfur found in the particle phase. The measured MSA : sulfate ratio ($\sim 2.5 : 1$) is in broad agreement with those reported in Chen et al. (2012). Minor species observed included dimethyl sulfoxide (DMSO), $C_2H_6SO_2$ (likely dimethyl sulfone, $DMSO_2$), and methane sulfinic acid (MSIA), known products from the addition channel, as well as $CH_2SO_2$ (likely a thioacid, which may be formed as an OH oxidation product of HPMTF, Jernigan et al., 2022a) and $CH_3SO_6N$ (likely methanesulfonyl peroxynitrate, formed from $CH_3S(O)_2OO + NO_2$). No HPMTF was observed in these experiments, which is expected given the short bimolecular $RO_2$ lifetime.

In Exp. 2a (low-NO conditions, Fig. 2b), $H_2O_2$ was the OH precursor, and NO and $HO_2$ levels were sufficiently low ($\sim 10$ and 100 ppt, respectively) enough for $RO_2$ isomerization to dominate ($\tau_{bi} > 10$ s). $HO_2$ generated from $H_2O_2 + OH$ is expected to promote the formation of $CH_3SCH_2OOH$ from Reaction (R2); however, we cannot distinguish $CH_3SCH_2OOH$ from its isomer, $DMSO_2$. Product distributions are dramatically different than those under high-NO conditions. The total sulfur products measured accounted for nearly all (90 % (64 %–118 %)) of the reacted DMS sulfur; this sulfur closure is good but slightly worse than in Exp. 1. The larger uncertainty range is due to the uncertainty of the HPMTF calibration in the $I^-$ CIMS. However, the near-sulfur closure, derived from multiple independently calibrated instruments, combined with the HPMTF yields (discussed in Sect. 3.3) suggest that our estimated sensitivity is reasonably accurate, and thus our overall uncertainty of total sulfur may be an overestimate.

Due to the long $RO_2$ bimolecular lifetime ($\tau_{bi} > 10$ s), the dominant product is HPMTF from $CH_3SCH_2OO$ isomerization; this accounts for about half of the reacted sulfur (60 % of the measured product sulfur). It is expected that a negligible amount (1 % or less) of HPMTF was lost to the chamber walls under the experimental condition here based on its estimated vapor pressure (see the Supplement). The time series of $C_2H_4SO_3$-$^{12}C_2$ in the $I^-$ CIMS ($C_2H_4SO_3 \cdot I^-$) and in the $NH_4^+$ CIMS ($C_2H_4SO_3 \cdot NH_4^+$), shown in Fig. S2, match very well. This indicates that there was negligible $N_2O_5$ formation from the residual $NO_x$ in the chamber, since $N_2O_5$ is

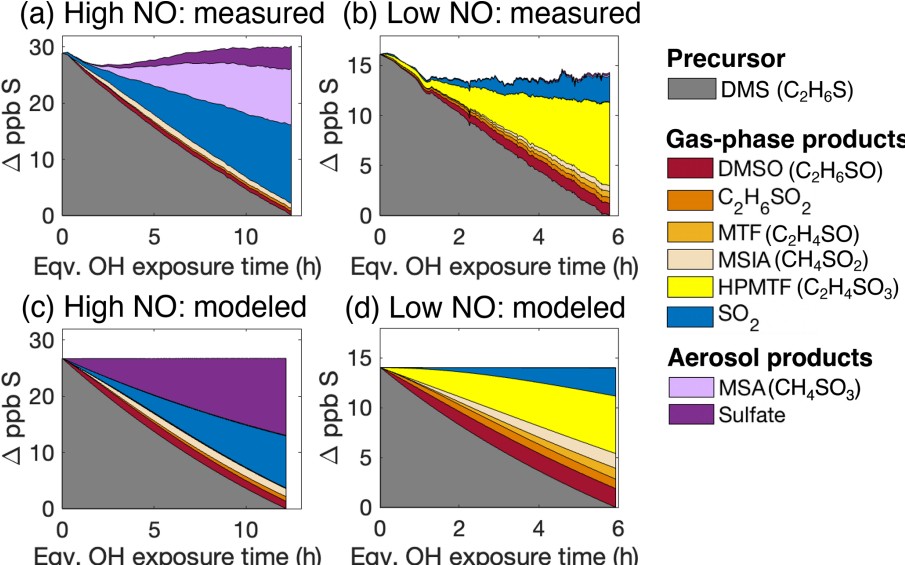

**Figure 2.** Stacked plots showing the total sulfur measured (**a** and **b**) and modeled (**c** and **d**) under high-NO (**a** and **c**) and low-NO (**b** and **d**) conditions. Panels (**a**) and (**c**) are for Exp. 1, and panels (**b**) and (**d**) are for Exp. 2a. Data shown in panel (**b**) are from DMS-$^{12}C_2$ and DMS-$^{13}C_2$ combined. Products with a formula of $C_2H_6SO_2$ may be $DMSO_2$ and/or $CH_3SCH_2OOH$; under high-NO conditions, they are likely to be predominantly $DMSO_2$. Minor products detected but not listed in the legend due to their very low concentrations include $CH_2SO_2$ (a sulfene or thioacid) and $CH_3SO_6N$ (likely methanesulfonyl peroxynitrate). Note that $y$ axes denote the changes in concentrations of the precursor and products.

not measurable by the $NH_4^+$ CIMS, and therefore our quantification of HPMTF-$^{12}C_2$ in Exp. 2a with $I^-$ CIMS is free of $N_2O_5$ interferences. Only 3.3 % (3.1 %–5.4 %) of the reacted sulfur was found in the aerosol by the end of the experiment.

## 3.2   Model–measurement comparison

The (near-)sulfur closure of the experiments, in which virtually all the reacted sulfur was measured as products, enables a comparison with the mechanistic model. MCM predictions for the two experiments described above (Exp. 1 and 2a) are shown in Fig. 2c and d; individual species are also compared in Figs. S3 and S4. Under high-NO conditions, measurements and model predictions (Figs. 2a and c, S3) agree well for gas-phase species and for total particulate sulfur. However, the two differ greatly in terms of particle-phase composition: AMS measurements indicate $\sim 70$ % of the particle-phase sulfur is MSA, with the remainder as sulfate; by contrast, the model predicts that sulfate dominates, with a negligible ($\sim 0.1$ %) contribution from MSA. This suggests the mechanism may underestimate the rate of MSA formation (a result consistent with recent studies; Wolleson de Jonge et al., 2021; Shen et al., 2022) and/or overestimate the rate of sulfuric acid formation.

In the MCM, both MSA and sulfuric acid are formed from reactions of the $CH_3S(O)_2O$ radical[TS1]:

$$CH_3S(O)_2O + HO_2 \rightarrow CH_3S(O)_2OH(MSA) + O_2 \quad \text{(R4)}$$

$$CH_3S(O)_2O + M \rightarrow CH_3 + SO_3. \quad \text{(R5)}$$

Reaction (R5) generates sulfur trioxide ($SO_3$), which will quickly hydrolyze to form sulfuric acid. $SO_3$ can also be formed by the OH oxidation of $SO_2$, but this reaction would occur over 50 h of OH exposure, much longer than the oxidation timescale in Exp. 1. Since the measured and modeled total particulate sulfur (MSA + sulfate) agree well, the model–measurement differences in the ratio of MSA to sulfuric acid (or sulfate) may relate to the relative rates of these $CH_3S(O)_2O$ reactions. It is possible that the rate constant of Reaction (R4) is underestimated in the mechanisms, but even if it is increased it to a gas-kinetic rate ($3 \times 10^{-10}$ cm³ molec.$^{-1}$ s$^{-1}$), MSA is still not predicted to dominate over sulfuric acid. Instead, the decomposition of $CH_3S(O)_2O$ (Reaction R5), which has received little study, might be slower than the value used in the mechanism ($\sim 0.09$ s$^{-1}$), leading to slower sulfuric acid formation. Alternatively, MSA might be formed by the reaction of $CH_3S(O)_2O$ with species other than $HO_2$, such as DMS or HCHO (Barnes et al., 2006; Yin et al., 1990). While such reactions are unlikely to be important in the atmosphere, they might occur in laboratory experiments, which have relatively high concentrations of organic species. However, the kinetics of such reactions are not well known and warrant future research.

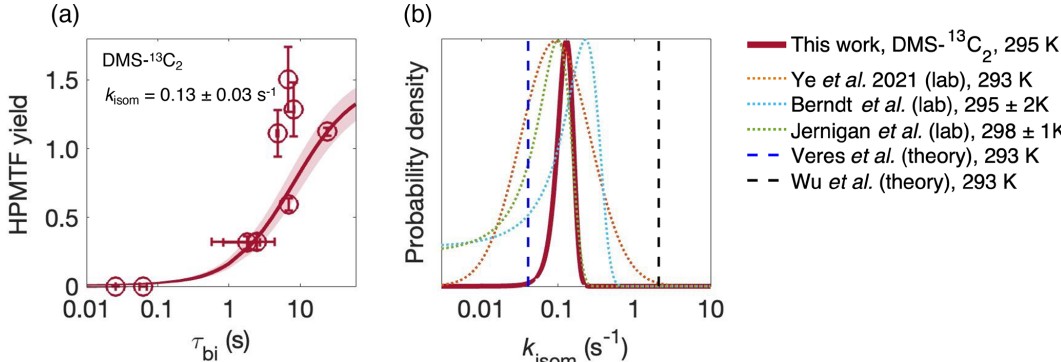

**Figure 3. (a)** The yield of HPMTF in the abstraction channel as a function of the bimolecular lifetime $\tau_{bi}$ of $CH_3SCH_2OO$ from the DMS-$^{13}C_2$ data. The shaded area is $1\sigma$ of the fit, which takes into account uncertainty in both $\tau_{bi}$ (arising from errors in [NO] and [$HO_2$]) on the $x$ axis, and instrument noise on the $y$ axis. Uncertainty in the CIMS sensitivity to HPMTF affects the absolute measurements but not the inflection point of the curve or the derived value of $k_{isom}$. **(b)** Comparison of $k_{isom}$ from this work with previous determinations of $k_{isom}$ at 293–298 K (Berndt et al., 2019; Jernigan et al., 2022a; Veres et al., 2020; Wu et al., 2015; Ye et al., 2021).

Another potential source of MSA is the oxidation of MSIA by OH (Yin et al., 1990; Lucas and Prinn, 2002; von Glasow and Crutzen, 2004; Wollesen de Jonge et al., 2021; Shen et al., 2022). This pathway is currently not included in the MCM, which has MSIA reacting with OH to form $SO_2$ and $CH_3$ (Fig. 1). It has been suggested (Yin et al., 1990) that the reaction may occur via abstraction of the acidic hydrogen: TS2

$$CH_3S(O)OH(MSIA) + OH \rightarrow CH_3S(O)O + H_2O. \qquad (R6)$$

As shown in Fig. 1, the resulting $CH_3S(O)O$ radical may react with ozone to form $CH_3S(O)_2O$, which can react further to form MSA or $SO_3$ (Reactions R4–R5). However, inclusion of this reaction in the model increases MSA formation only slightly, and the model–measurement discrepancy remains large (Fig. S5). Alternatively, OH might add to MSIA (Lucas and Prinn, 2002; Arsene et al., 2002; Shen et al., 2022), forming the intermediate $CH_3SO(OH)_2$ that can react with $O_2$ to produce MSA: TS3

$$CH_3S(O)OH(MSIA) + OH \xrightarrow{M} CH_3S(O)(OH)_2 \qquad (R7)$$
$$CH_3S(O)(OH)_2 + O_2 \rightarrow CH_3S(O)_2(OH)(MSA) + HO_2. \qquad (R8)$$

Including these reactions into the mechanism, using the rate constant for MSIA + OH suggested by the MCM ($9 \times 10^{-11}\ \mathrm{cm^3\ molec.^{-1}\ s^{-1}}$) substantially increases the predicted MSA but at the same time decreases the predicted $SO_2$ concentration, worsening the model–measurement agreement for $SO_2$, and does not change predicted sulfate formation, leading to an overestimate in total aerosol production (Fig. S5). Taken together, while the OH oxidation of MSIA (Reactions R6–R8) may contribute to MSA formation, it appears not to be the only (or major) source for the MSA model–measurement discrepancy in the present experiments.

In the low-NO case (Figs. 2b and d, S4), measured and modeled concentrations also broadly agree. The predicted concentration of HPMTF is lower (by $\sim 30\%$) than what was measured. This could be due to the uncertainty in the sensitivity of HPMTF in the $I^-$ CIMS and/or in the $k_{isom}$ value used in the model. The $k_{isom}$ value used, $0.09\ \mathrm{s^{-1}}$, is derived from our previous study (Ye et al., 2021); as discussed below, this value agrees with that determined in this work. Compared to measurements, the model also predicts somewhat higher concentrations of minor sulfur-containing products, such as DMSO, $C_2H_6SO_2$ ($DMSO_2 + CH_3SCH_2OOH$), MSIA, and MTF. This could be caused by overestimates of instruments' sensitivities, uncertainties in the rate constants in the model, or some losses to surfaces. Nevertheless, overall the model and measurements agree quite well, with product formation dominated by HPMTF and little aerosol formation since low-volatility species (MSA and sulfuric acid) are formed only as later-generation products.

## 3.3 Determination of $k_{isom}$

The fate of the $CH_3SCH_2OO$ radical, and hence the product distribution of DMS oxidation, relies critically on the isomerization rate constant of the $CH_3SCH_2OO$ radical ($k_{isom}$). In our previous work we determined $k_{isom}$ from a single reaction condition (at one value of $\tau_{bi}$), and the $k_{isom}$ value had a large uncertainty due to the poorly constrained sensitivity of HPMTF in the CIMS. Here, we determine $k_{isom}$ by examining product formation at multiple values of $\tau_{bi}$, similar to previous measurements of isomerization rates of terpene-derived $RO_2$ radicals (Xu et al., 2019). MCM modeling suggests that $RO_2 + RO_2$ reactions represent $\sim 1\%$ of the $RO_2$ sink in the experiments, and therefore the only bimolecular reactions considered are $RO_2 + NO$ and $RO_2 + HO_2$. HONO or NO was added to the chamber several times during the experiment (Fig. S6), perturbing the branching of the $CH_3SCH_2OO$ radical (isomeriza-

tion vs. bimolecular reactions). The total S measurements are shown in Fig. S8. The yield of HPMTF in the abstraction channel ($\Delta$[HPMTF] / $\Delta$[DMS]$_{\text{abs}}$) was calculated for each perturbation as a function of $\tau_{\text{bi}}$ after taking into account the of loss via OH oxidation ($k_{\text{OH+HPMTF}} = 2.1 \times 10^{-11}$ cm$^3$ molec.$^{-1}$ s$^{-1}$; see Sect. 3.4). The detailed calculation is described in the Supplement (Eqs. S1–S4). Figure 3a shows the HPMTF yield as a function of $\tau_{\text{bi}}$. As expected, the yield increases dramatically with $\tau_{\text{bi}}$, and fitting these data to Eq. S4 (given in the Supplement) enables the determination of $k_{\text{isom}}$. The best-fit value for $k_{\text{isom}}$ is $0.13 \pm 0.03$ s$^{-1}$. The uncertainty is much smaller than in our previous determination (Ye et al., 2021) since the fit depends only on the shape (the inflection point) of the curve and not the absolute yield values and thus is insensitive to the uncertain HPMTF calibration factor. Nonetheless, since the asymptotic (high $\tau_{\text{bi}}$) value is close to 1 (1.5), our estimated calibration factor appears to be reasonably accurate. The three data points with higher HPMTF yields (top of Fig. 3a) were collected in the latter half of the experiment, after HPMTF had built up in the chamber, and therefore correcting for OH loss resulted in an increased HPMTF yield. Because of their larger measurement uncertainties, these data points have smaller effects on the overall fit to Eq. (S4). If the OH loss is not included, $k_{\text{isom}} = 0.11 \pm 0.02$ s$^{-1}$ (Fig. S7).

Figure 3b compares our value of $k_{\text{isom}}$ with previous measurements and theoretical determinations ($T = 293$–298 K) (Berndt et al., 2019; Jernigan et al., 2022a; Veres et al., 2020; Wu et al., 2015; Ye et al., 2021). Our measured value of $k_{\text{isom}}$ is consistent with our previous (single $\tau_{\text{bi}}$) measurement (Ye et al., 2021) though with a much reduced uncertainty and is also in broad agreement with measured values from Berndt et al. (2019) ($0.23 \pm 0.12$ s$^{-1}$) and Jernigan et al. (2022) ($0.1 \pm 0.05$ s$^{-1}$).

## 3.4   Reaction rates of OH with HPMTF and MTF

Here we examine the oxidation of HPMTF and MTF, two species whose chemical fates are not well known. Both were formed only under low-NO conditions (Exp. 2a); because of the relatively low OH concentrations of that experiment, their concentrations increased throughout the entire experiment, with no subsequent decay. Thus, to estimate $k_{\text{OH+HPMTF}}$ and $k_{\text{OH+MTF}}$, high concentrations of NO ($\sim 70$ ppb) were introduced at the end of Experiment 2 (denoted as Exp. 2b, shown in Fig. 4a). The large amount of NO essentially terminated the production of HPMTF and MTF and at the same time increased the OH concentration in the chamber. The total sulfur product distribution for Exp. 2 is shown in Fig. S9. The loss of HPMTF during this period, shown in Fig. 4b, is expected to be dominated by OH reaction because the high level of NO precluded substantial oxidation by O$_3$ and NO$_3$. Photolysis of HPMTF is also unlikely to contribute to the observed decay: by assuming that its photolytic cross sections are equal to the summed cross section of aldehydes and organic perox-

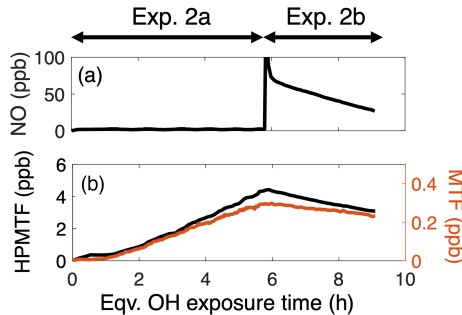

**Figure 4. (a)** NO concentration measured by the NO–NO$_2$–NO$_x$ analyzer in Exp. 2. At OH exposure $\sim 5.8$ h, 70 ppb of NO was injected into the chamber. **(b)** Time series of HPMTF-$^{12}$C$_2$ and MTF-$^{12}$C$_2$ in Exp. 2. The decay of HPMTF and MTF was used to estimate their reaction rate coefficients with OH.

ides taken from MCM (Khan et al., 2021), we estimate that photolysis accounted for only 4 % of the HPMTF loss in our chamber. Using the cross section for MTF measured by Patroescu et al. (1996), we obtain an even lower photolysis rate, with photolysis accounting for less than 2 % of HPMTF loss in the chamber.

By calculating [OH] using the decay of DMS after the addition of NO, we fit the decay of HPMTF (Figs. 4b and S10) to derive $k_{\text{OH+HPMTF}}$ of 2.1 (2.0–2.2) $\times 10^{-11}$ cm$^3$ molec.$^{-1}$ s$^{-1}$. This is in agreement with recent measurements of Jernigan et al. (2022) (1.4 (0.27–2.4) $\times 10^{-11}$ cm$^3$ molec.$^{-1}$ s$^{-1}$); both experimental values are an order of magnitude higher than an earlier theoretical estimate of the rate ($1.2 \times 10^{-12}$ cm$^3$ molec.$^{-1}$ s$^{-1}$) (Wu et al., 2015). Using this lower value, Khan et al. (2021) estimated that photolysis loss dominates HPMTF sink in the global marine sulfur budget, with OH oxidation only accounting for 10 % of HPMTF loss. This higher OH rate constant suggests that OH oxidation is in fact likely to be an important loss process for HPMTF, at least when liquid water is not present (Fung et al., 2022; Vermeuel et al., 2020; Novak et al., 2021).

MTF is formed predominantly as a second-generation DMS oxidation product from CH$_3$SCH$_2$OOH + OH in low-NO conditions in the mechanism. Using a similar method as $k_{\text{OH+HPMTF}}$ (Figs. 4b and S10), $k_{\text{OH+MTF}}$ is estimated to be 1.35 (1.3–1.4) $\times 10^{-11}$ cm$^3$ molec.$^{-1}$ s$^{-1}$, which agrees with the only other measurement of $k_{\text{OH+MTF}}$, $1.11 \pm 0.22 \times 10^{-11}$ cm$^3$ molec.$^{-1}$ s$^{-1}$, by Patroescu et al. (1996).

## 3.5   Role of relative humidity

The experiments described above were carried out under dry conditions and thus focus only on homogenous gas-phase chemistry; in the atmosphere, heterogeneous and aqueous-phase processes may also be important contributors to DMS oxidation chemistry (Hoffmann et al., 2016). Thus, Experiments 4 and 5 were carried out at 65 % RH, under high-

and low-NO levels, respectively. These experiments were carried out over longer timescales (higher OH exposures) than the corresponding dry experiments to better probe multigenerational product formation.

Results from Exp. 4 (in which 50–100 ppb NO was maintained in the chamber) are shown in Fig. 5a. The overall product distribution is similar to that under dry conditions (Fig. 2a), with $SO_2$, MSA, and sulfate being the major reaction products. The modeled product distribution shown in Fig. S11a is largely the same as that in the dry experiment (Fig. 2c), as water does not play a role in the gas-phase oxidation mechanism shown in Fig. 1. Even though this experiment was carried out over longer timescales, the measured sulfur closure is quite good, 107 % (99 %–171 %) of the reacted DMS at the end of the experiment.

Figure 5c compares the evolving concentrations of major product species under high- and low-RH conditions, presented as change in product concentration relative to change in DMS concentration, over the initial OH exposure (corresponding to that of Exp. 1). Over these timescales, species such as DMSO, $SO_2$, and MSA showed a relatively small effect of RH. By contrast, almost no $C_2H_6SO_2$ (likely $DMSO_2$) was measured in the gas phase under high-RH conditions. Within the timescale of the experiments, our measurements do not suggest conversion of MSA to sulfate in the aerosol phase, as predicted in some modeling studies (Fung et al., 2022; Chen et al., 2018). This difference may arise from low particle-phase OH concentrations in our experiments.

Figure 5b shows products from Exp. 5 (65% RH, low-NO conditions, $\tau_{bi} > 1$ s). As in the low-RH, long-$\tau_{bi}$ case (Exp. 2a, Fig. 2b), HPMTF and $SO_2$ are the dominant measured products, and little aerosol formation is observed. One minor new product, with formula $SO_6$, was detected in the $I^-$ CIMS in this experiment; it is likely an adduct (i.e., $O_3 \cdot SO_3 \cdot I^-$) or a fragment formed in the instrument, but the parent species is unknown. In contrast to the high-NO experiment (Exp. 4), sulfur closure was markedly worse than under dry conditions. In the first 6 h of equivalent OH exposure (the timescale of the dry experiment), only 74 % (53 %–97%) of the reacted sulfur was detected as products. This sulfur closure degraded still further as the experiment proceeded and was only 23 % (18 %–31 %) at the end of the experiment. Here, $I^-$ CIMS sensitivities derived from the dry calibration were used for species quantification and therefore may underestimate the concentration under high-RH conditions (Lee et al., 2014; Veres et al., 2020). However, these differences would have to be dramatic (by factor of 5 or more) to account for all the reacted sulfur, and therefore such calibration errors are unlikely to explain the decreased sulfur closure.

Figure 5d shows differences for key product species formed in the long-$\tau_{bi}$ experiments under the high- and low-RH conditions, again over the timescales of the dry experiment (the first 6 h of equivalent OH exposure). Over these timescales, the initial yields of DMSO, $C_2H_6SO_2$, and

HPMTF are not substantially different in the humid and dry cases. $SO_2$ concentrations were lower under humid conditions but with an absolute difference of only $\sim 2$ ppb. Thus the production rates of these species are not affected dramatically by RH level. Instead the poor sulfur closure at high-RH conditions suggests that extra losses over longer timescales may be most likely by uptake to surfaces. The low aerosol concentration towards the end of the experiment (due to particle wall loss over the long experimental time, $\sim 17$ h) could lead to substantial chamber wall loss of low-volatility products, which would contribute to this gap in measured sulfur. Such surface losses are likely exacerbated at high-RH conditions, due to uptake into the aqueous phase. The initial aerosol liquid water content (LWC) in the high-RH experiment was 10–100 µg m$^3$, orders of magnitude lower than LWC in maritime clouds (Wallace and Hobbs, 2006). Therefore, such losses may play an even more important role in the real atmosphere. Indeed, studies have suggested that uptake to cloud water may be an important sink of gas-phase HPMTF. Using in situ measurements, Vermeuel et al. (2020) and Novak et al. (2021) have shown that HPMTF is lost to clouds and aerosols effectively in the marine boundary layer. Similarly, using a global model, Fung et al. (2022) found that including cloud uptake into a global model substantially decreases the global burden of HPMTF, by up to 86 %. This uptake of water-soluble intermediate species (e.g., MSIA, $DMSO_2$, and HPMTF) into cloud droplets may then contribute to the condensed-phase production of MSA and sulfate (Hoffmann et al., 2021; Novak et al., 2021), but such processes are not accessed in the present chamber experiment.

## 4 Conclusions

In this study, we conducted a series of chamber experiments to investigate the total product distribution from DMS oxidation at different $RO_2$ fates and relative humidities. Under dry conditions, good sulfur closure was obtained, suggesting most of the sulfur-containing product species were accounted for. Under high-NO conditions ($\tau_{bi} < 0.1$ s), major products are $SO_2$, MSA, and sulfate, whereas under low-NO conditions ($\tau_{bi} > 10$ s), HPMTF formed from $RO_2$ isomerization makes up about half of the product sulfur, with very little MSA or sulfate formation. Comparisons between measurements and MCM predictions show relatively good agreement for most species and total aerosol formation. However, under high-NO conditions, the model predicts much more sulfate and less MSA than was measured; this might indicate errors in the kinetics of the reactions that lead to rapid (first-generation) MSA or sulfate formation. This work also provides new measurements of the rate constants (at 295 K) of key reactions in the DMS oxidation mechanism, including $k_{isom}$ ($0.13 \pm 0.03$ s$^{-1}$), $k_{HPMTF+OH}$ ($2.1 \times 10^{-11}$ cm$^3$ molec.$^{-1}$ s$^{-1}$), and $k_{MTF+OH}$ ($1.35 \times 10^{-11}$ cm$^3$ molec.$^{-1}$ s$^{-1}$). Our measured

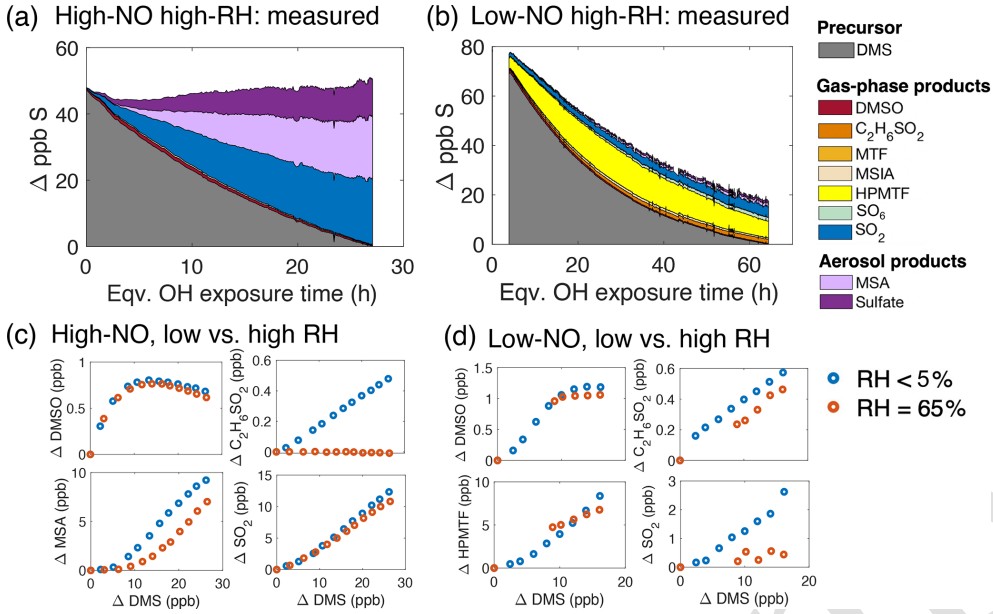

**Figure 5.** Results from the high-humidity (65 % RH) DMS oxidation experiments. **(a)** Product formation under high-NO conditions (Exp. 4). **(b)** Product formation under low-NO conditions (Exp. 5). Because of instrument downtime, no data were collected for the first 4 h of equivalent OH exposure. **(c)** Comparison of major species between the low-RH (Exp. 1) and high-RH experiment (Exp. 4) under high-NO conditions. **(d)** Comparison of major species between the low-RH (Exp. 2) and high-RH experiment (Exp. 5) under low-NO conditions. Changes in product concentrations are plotted against changes in DMS concentration over the initial 6 h of OH exposure, when data from both the dry and high-RH experiments were available.

value of $k_{\mathrm{HPMTF+OH}}$, which is consistent with that of Jernigan et al. (2022a), suggests that OH is a more important gas-phase sink of HPMTF than photolysis. Lastly, results from high-RH conditions suggest heterogeneous losses of at least some of the products, indicating that uptake into the atmospheric aqueous phase (e.g., cloud droplets) may be an important sink as well.

Taken together, our results show that RO$_2$ fate has a controlling influence on the distribution of sulfur-containing products from DMS oxidation. In particular, the formation of HPMTF from RO$_2$ isomerization suppresses (or at least delays) the gas-phase formation of SO$_2$, sulfate, and MSA. Additional studies are needed to constrain the temperature dependence of $k_{\mathrm{isom}}$ to predict the formation of HPMTF (and other products) in warmer or colder environments, as well as to characterize the full multiphase product distribution under higher-RH conditions. In addition, experiments carried out over longer oxidation timescales and with different oxidants are needed to better understand the amount and rate of aerosol formation over days of oxidation. A related need is improved constraints on the atmospheric fate of HPMTF and other key reaction intermediates (e.g., DMSO, MSIA), including rates and products of gas-phase oxidation, aqueous-phase oxidation, and photolysis, as well as rates of physical loss (deposition and uptake).

**Data availability.** Chamber data and species concentrations for all experiments and model outputs have been archived and are available via the Kroll Group publication website at http://krollgroup.mit.edu/publications.html (Kroll Group, 2022). The F0AM model used in this work is publicly available at https://github.com/AirChem/F0AM/releases/tag/v3.2 (Wolfe, 2022), and the latest release is available at https://zenodo.org/record/6984581 (Wolfe et al., 2022) TS4.

**Supplement.** The supplement related to this article is available online at: https://doi.org/10.5194/acp-22-1-2022-supplement.

**Author contributions.** QY, MBG, JEK, FM, AZ, YL, and JRR collected the data. QY and MBG analyzed the data. MBG performed box model simulations. QY and JHK wrote the manuscript. MC, FNK, CLH, and JHK provided project guidance. All authors were involved in helpful discussion and contributed to the manuscript.

**Competing interests.** At least one of the (co-)authors is a member of the editorial board of *Atmospheric Chemistry and Physics*. The peer-review process was guided by an independent editor, and the authors also have no other competing interests to declare.

**Disclaimer.** Publisher's note: Copernicus Publications remains neutral with regard to jurisdictional claims in published maps and institutional affiliations.

**Acknowledgements.** The authors thank Timothy Bertram, Gordon Novak, and Chris Jernigan at the University of Wisconsin–Madison for insightful discussions.

**Financial support.** This work was supported by the U.S. Department of Energy Biological and Environmental Research program (grant no. DE-SC0018934) and the Harvard Global Institute.

**Review statement.** This paper was edited by Sergey A. Nizkorodov and reviewed by three anonymous referees.

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

## Remarks from the language copy-editor

CE1     Please note the hyphenation in line with our standards.

CE2     Please note the changes here.

## Remarks from the typesetter

TS1     Due to the requested changes (removal of MSA), we have to forward your requests to the handling editor for approval. To explain the corrections needed to the editor, please send me the reason why these corrections are necessary.

TS2     Due to the requested changes (removal of MSIA), we have to forward your requests to the handling editor for approval. To explain the corrections needed to the editor, please send me the reason why these corrections are necessary.

TS3     Due to the requested changes (removal of MSIA), we have to forward your requests to the handling editor for approval. To explain the corrections needed to the editor, please send me the reason why these corrections are necessary.

TS4     Please note added references in this section.

TS5     Please adjust reference if necessary and provide date of last access.

TS6     Please adjust reference if necessary and provide date of last access.

TS7     Is code correct? If so, we could change the section "Data availability" to "Code and data availability" section.

TS8     Please note added reference.