# Peer review of "Supporting Information for"

_Atmospheric Chemistry and Physics, 2022_

## Referee Comment (RC3)

Ms.No. acp-2022-566

The authors describe experimental findings from the OH-initiated oxidation of DMS conducted in a chamber under almost atmospheric conditions. Bimolecular $RO_2$ lifetime was varied in a wide range from > 10 s to < 0.1 s by changing NO in the reaction gas from 10 ppt up to 50 ppb. A suite of analytical techniques was used for comprehensive analysis of the resulting product distribution in the gas phase as well as of the formed aerosol products on a seed. Reaction conditions for the long bimolecular lifetime, standing for the pristine atmosphere, allowed studying the product formation in the abstraction channel governed by $CH_3SCH_2O_2$ isomerization. The rate coefficient of this isomerization step was re-investigated being in good agreement with other experimental values. Especially the fantastic sulfur closure in the measured products is worth mentioning, which demonstrates the accuracy in conducting the experiment and in product analysis.
This work represents a next, very nice piece of work from this group on the understanding of chemical processes in the reaction of OH with DMS, here especially for the 1st generation gas-phase products.
The manuscript is well written and suitable for publication in this journal. Here only a few minor comments:

1) While the agreement between experiment and model in the low-NO case is very good, there are clear differences in the high-NO case. Here, the MSA production seems to be strongly underestimated by the mechanism. High-NO experiments have been already done 20 years before by the Wuppertal group and by others, mostly with relatively high reactant concentrations. Are the findings from the present work consistent with the former results? Can we learn anything from the comparison?

2) Product distributions are presented for very long and very low bimolecular $RO_2$ lifetimes, > 10 s and < 0.1 s. But what happens in between? The $RO_2$ lifetime was varied in order to estimate the rate coefficient of $CH_3SCH_2O_2$ isomerization based on HPMTF yields. Could the authors provide more information on the other products as a function of lifetime?

3) k($CH_3SCH_2O_2$ isomerization) was obtained in an indirect way relative to k($RO_2$ + NO), right? What was the used value of k($RO_2$ + NO)? It´s important to have a comparison with other studies. Was the $CH_3SCH_2O_2$ + $HO_2$ reaction neglected in the analysis? Please explain more in detail the determination of k(isomerization).
Moreover, $CH_3SCH_2O_2$ isomerization leads to the next $RO_2$ radical $HOOCH_2SCH_2O_2$ after $O_2$ addition. And the next isomerization step in $HOOCH_2SCH_2O_2$ finally ends up in HPMTF + OH. What is known about the rate of $HOOCH_2SCH_2O_2$ isomerization? Is $HOOCH_2SCH_2O_2$ + NO distinctly slower even for the high NO addition of 50 ppb? Otherwise it must be considered in the data analysis because HPMTF was taken for the determination of k($CH_3SCH_2O_2$ isomerization), not the direct isomerization product $HOOCH_2SCH_2O_2$.

4) It is not the first time that HPMTF was detected by ammonium CIMS. In Berndt et. al a suite of ionization schemes was used for HPMTF monitoring including ammonium (with the PTR3), see fig. S3 and explanation in the main body.

---

## Author Comment (AC1)

Reviewer 1:
Summary:

This manuscript describes a system of environmental chamber experiments to examine the kinetics and product distribution both in the gas and aerosol phase of DMS oxidation under a variety of oxidative and environmental conditions. The work progresses the field by providing a stronger constraint on a gas-phase production and loss pathway of a recently discovered sulfur oxidation product, hydroperoxymethyl thioformate (HPMTF). The authors present an isomerization rate constant that fits within existing literature values as well as a constrained OH loss rate with a method for calculation validated by the co-observation of the OH loss rate of a previously calculated sulfur species, methyl thioformate. The work adds to the field through a reexamination of the MSA to sulfate yield within their chamber experiments under differing relative humidity and NO concentrations. They propose that future work focus on better constraining the oxidation mechanism leading to the formation of MSA and sulfate. The manuscript reads well and the results are robust. The manuscript should be published after the following comments have been addressed.

We thank the reviewer for the positive comments and the valuable suggestions.

General comments:

The manuscript focuses on the gas-phase oxidation pathways of DMS with a focus on HPMTF. The chamber was run under a variety of different oxidative environments that drastically change the RO2 lifetime with the inclusion of NO. I recommend adding a minor discussion and reminder of the reactions that could occur in the chamber under the various oxidative conditions that may be amplified compared to those occurring in typical marine boundary layer conditions. In particular, including the H2O2 + OH reaction and its ability to form HO2 a reactant with methylthiomethyl peroxy radical (MTMP) as a competing reaction with isomerization. A review will help the reader understand the experiments and how the oxidative conditions were set up.

We thank the reviewer for the suggestion. In Line 53 – 55, we listed all the major reactions of the $CH_3SCH_2OO$ including the reaction with $HO_2$.

In the revised manuscript, in Section 3.1, we have added the following text in Line 181:

$HO_2$ generated from $H_2O_2$ + OH is expected to promote the formation of $CH_3SCH_2OOH$ from Reaction (2), however, we cannot distinguish $CH_3SCH_2OOH$ from its isomer, $DMSO_2$."

There is a discussion on the role of MSA formation and its relationship to sulfate. The focus is on the gas-phase mechanisms that yield MSA and H2SO4. I recommend the authors comment on the role of the heterogeneous and aqueous processing of the sulfur compounds and their tie to MSA and sulfate formation. In particular, the ability for water soluble species (i.e. HPMTF, MSIA, DMSO, DMSO2) to condense onto available aerosol surface and contribute to sulfate or MSA formation. The authors utilize a variety of seed particles without any reference in the main text or SI to their impact on the fate of the sulfur molecules. Is there future work in preparation or was there no observable effect on the seed composition?

We thank the reviewer for the comment. We used non-sulfate seeds to prevent any interference from seed particles on quantifying secondary sulfate yield. We think the seeds themselves under dry conditions and the NaCl seeds in the high-RH experiment should be quite inert. In terms of aqueous processing that convert intermediate species to MSA and sulfate, one major challenge is that the oxidant concentration in the aqueous phase is difficult to constrain. Exploring the effects of seed particle composition on aerosol-phase processes is an important future direction (such as in Jernigan et al, 2022) but is beyond the scope of this study.

In the revised manuscript, we added more discussions on the effects of seeds and the heterogeneous processes that may affect product distribution.

In Line 348, we have added:

"This uptake of water-soluble intermediate species (e.g., MSIA, $DMSO_2$ and HPTMF) into cloud droplets may then contribute to the condensed-phase production of MSA and sulfate (Hoffmann et al., 2021; Novak et al., 2021) but such processes are not accessed in the present chamber experiment."

In Line 121 we have added:

"In these experiments, non-sulfate seeds were used to avoid interferences when quantifying secondary sulfate in the aerosols. For low-RH experiments (Exp. 1-3), ammonium nitrate seed particles were used, since dry ammonium nitrate particles are expected to be chemically inert. For the high-RH high NO experiment (Exp. 4), NaCl particles were used. As discussed below, major products are similar to those in the high-NO dry experiment, suggesting that the NaCl seed particles in Exp. 4 have little to no effect on the product distribution in these experiments. More studies are needed to constrain the effects of different seed particle on the reactive uptake of DMS oxidation products (Jernigan et al., 2022b)."

I recommend citing additional literature on the previous work looking at the MSA and sulfate yields when higher concentrations of NOx are present (Chen et al 2012, www.atmos-chem-phys.net/12/10257/2012/, Patroescu et al 1999, https://doi.org/10.1016/S1352-2310(98)00120-4). I would recommend making connections between these previous chamber reports and this current work to see if connections and chemical pathways can be made.

We thank the reviewer for the suggestion. In the high-NO high-RH experiment in Chen et al (2012) (similar experimental condition in Experiment 4 in our paper), the sulfur yield for sulfuric acid are lower than yields in our work. However, it is unclear whether they reported wall loss-corrected aerosol concentration in all of their experiments. Despite the lower yields, the MSA:sulfate in our work is broadly agreement with that in Chen et al (MSA:sulfate ~ 3).

We also extended the discussions on MSA formation, incorporating mechanisms proposed in previous studies into the model to investigate MSA model-measurement comparison.

In the revised manuscript, we have included the following text to better connect our work with other studies:

In Line 173:

"The measured MSA:sulfate ratio (~2.5:1) is in broad agreement with those reported in Chen et al. (2012)."

In Line 223, we have added the following paragraph:

"Another potential source of MSA is the OH-initiated oxidation of MSIA by OH (Yin et al., 1990; Lucas and Prinn, 2002; von Glasow and Crutzen, 2004; Wollesen de Jonge et al., 2021; Shen et al., 2022). This pathway is currently not included in the MCM, which has MSIA reacting with OH to form $SO_2$ and $CH_3$ (Figure 1). It has been suggested (Yin et al., 1990) that the reaction may occur via abstraction of the acidic hydrogen:
$$CH_3S(O)OH \text{ (MSIA)} + OH \rightarrow CH_3S(O)O + H_2O \tag{6}$$
As shown in Figure 1, the resulting $CH_3S(O)O$ radical may react with ozone to form $CH_3S(O)_2O$, which can react further to form MSA or $SO_3$ (reactions 4-5). However, inclusion of this reaction in the model increases MSA formation only slightly, and the model-measurement discrepancy remains large (Figure S5). Alternatively, OH might add to MSIA (Lucas and Prinn 2002; Arsene et al., 2002), forming the intermediate $CH_3SO(OH)_2$ that can react with $O_2$ to produce MSA:
$$CH_3S(O)OH \text{ (MSIA)} + OH \xrightarrow{M} CH_3S(O)(OH)_2 \tag{7}$$
$$CH_3S(O)(OH)_2 + O_2 \rightarrow CH_3S(O)_2(OH) \text{ (MSA)} + HO_2 \tag{8}$$
Including these reactions into the mechanism, using the rate constant for MSIA + OH suggested by the MCM ($9 \times 10^{-11}$ $cm^3$ $molec^{-1}$ $s^{-1}$) substantially increases the predicted MSA, but at the same time decreases the predicted $SO_2$ concentration, worsening the model-measurement agreement for $SO_2$, and does not change predicted sulfate formation, leading to an overestimate in total aerosol production (Figure S5). Taken together, while the OH oxidation of MSIA (reactions 6-8) may contribute to MSA formation, it is not the only (or major) source for the MSA model-measurement discrepancy in the present experiments."

References:
Chen, T., and M. Jang. "Chamber simulation of photooxidation of dimethyl sulfide and isoprene in the presence of NOx." Atmospheric Chemistry and Physics 12.21 (2012): 10257-10269.

Yin, Fangdon, et al. "Photooxidation of dimethyl sulfide and dimethyl disulfide. II: Mechanism evaluation." *Journal of Atmospheric chemistry* 11(4)365-399, 1990.

Von Glasow, R., and P. J. Crutzen. "Model study of multiphase DMS oxidation with a focus on halogens." *Atmospheric Chemistry and Physics* 4(3) 589-608, 2004

Wollesen de Jonge, Robin, et al. "Secondary aerosol formation from dimethyl sulfide–improved mechanistic understanding based on smog chamber experiments and modelling." *Atmospheric Chemistry and Physics* 21(13) 9955-9976, 2021

Shen, Jiali, et al. "High Gas-Phase Methanesulfonic Acid Production in the OH-Initiated Oxidation of Dimethyl Sulfide at Low Temperatures." *Environmental science & technology* (2022).

Lucas, D. D. and Prinn, R. G.: Mechanistic studies of dimethylsulfide oxidation products using an observationally constrained model. Journal of Geophysical Research: Atmospheres, 107(D14), ACH-12, 2002.

Arsene, C., Barnes, I., Becker, K. H., Schneider, W. F., Wallington, T. T., Mihalopoulos, N. and Patroescu-Klotz, I. V.: Formation of methane sulfinic acid in the gas-phase OH-radical initiated oxidation of dimethyl sulfoxide, Environ. Sci. Technol, 36(23), 5155–5163, 2002.

Lastly, a major advancement is the tighter constraint on the HPMTF + OH rate constant. I would recommend moving the S5 figure showing this result to the main text.

In the revised manuscript, we included a new figure (Figure 4).

Technical comments:

Line 56: I recommend a further discussion of the importance of RO2 reactions with other RO2 species present in the chamber. A comment is made about insignificance in the atmosphere, but this reaction pathway could be significant in the chamber. In addition, methyl thioformate (MTF) is thought to form through the reaction of MTMP with HO2, O2, OH or other RO2 species. The only channel explained later is through the OH oxidation of the product of MTMP with HO2 (CH3SCH2OOH). Could you please elaborate on the other potential chamber specific reactions here and why they were not addressed?

Based on our box modeling, $RO_2 + RO_2$ only represents ~1% of the $RO_2$ sink in the chamber. In the mechanism, MTF is formed via $CH_3SCH_2OOH + OH$ and via $CH_3SCH_2OO + RO_2$, and the $CH_3SCH_2OOH + OH$ is expected to be the major pathway.

In the revised manuscript, in line 255, we have added:

"MCM modelling suggests that $RO_2 + RO_2$ reactions represent ~ 1% of the $RO_2$ sink in the experiments, and therefore the only bimolecular reactions considered are $RO_2 + NO$ and $RO_2 + HO_2$."

Line 58: "believed to rapidly form SO2, sulfate, and methanesulfonic acid (MSA)" is slightly misleading. I would provide more clarification on this point. Most climate models have the H-abstraction lead to only SO2 while the OH addition leads to MSA and some SO2. I would specify that the rapid formation of MSA from this channel is only prominent under high HO2 and NOx conditions atypical of the marine environment.

We have changed the sentence (in Line 56) to:

"The $CH_3SCH_2O$ radicals formed from the NO pathway (Reaction 1) form $SO_2$, sulfate, and methanesulfonic acid (MSA) (Barnes et al., 2006)."

Line 64: Jernigan et al GRL 2022 also provided an isomerization rate constant, highlighted later but not here.

We have added Jernigan et al (2022).

Line 106: You cite two authors that found the MSA fragment (CH3SO2+) in the AMS is unique to MSA. Do these citations address the potential for DMSO2 to contribute to the CH3SO2+ fragment? There is significant discussion on DMSO2, so I wounder if you can kick off a CH3 from DMSO to make CH3SO2+.

This is something we wondered about ourselves. In Van Rooy (2019), the author atomized $DMSO_2$ into the AMS and failed to see any aerosol signal. We tried the same thing in our lab and likewise saw no aerosol signal. This is probably because $DMSO_2$ is volatile to semi-volatile (Scholz 2022) and therefore evaporated when entering the chamber. As a result, no aerosol-phase reference spectrum of $DMSO_2$ was obtained. It is expected that at least in the dry experiments, $DMSO_2$ stayed in the gas-phase and did not contribute to the $CH3SO_2^+$ fragment in the AMS.

In Section 2.5 in the SI, we have added:

"Particles containing $DMSO_2$ were also generated by atomizing a $DMSO_2$ solution and was directly introduced to the AMS. However, no aerosol signals were observed. This is probably because $DMSO_2$ is volatile to semi-volatile (Scholz et al., 2022) and therefore evaporated when entering the chamber. As a result, no aerosol-phase reference spectrum of $DMSO_2$ was obtained. It is expected that at least in the dry experiments, $DMSO_2$ stayed in the gas-phase and did not contribute to the $CH_3SO_2^+$ fragment in the AMS."

Reference:
Van Rooy, P. S. (2019). Secondary Aerosol Formation from the Oxidation of Amines and Reduced Sulfur Compounds. University of California, Riverside.

Scholz, Wiebke, et al. "Measurement Report: Long-range transport and fate of DMS-oxidation products in the free troposphere derived from observations at the high-altitude research station Chacaltaya (5240 m asl) in the Bolivian Andes." EGUsphere (2022): 1-42.

Line 115: How were the atomized seed particles added to the chamber? Were the particles dried before introduction to the chamber or were they added wet? I would recommend clarifying the

phase state of the seed particles added to the chamber under the high RH conditions. Line 297 starts to addresses this, but additional clarification would be greatly appreciated.

For dry experiments, the seed particles were dried using a diffusion dryer before entering the chamber. For high-RH experiments, the seed particles were introduced into the chamber directly from the atomizer without drying. The NaCl seeds in Experiment 4 were expected to be aqueous as 65% RH is below its efflorescence RH. The nitrate seeds in experiment 5 were also expected to be aqueous.

In Line 116, we have modified the text to:

"In dry experiments, seed particles (ammonium nitrate) were added into the chamber via first atomization followed by drying, providing surface area for condensing vapors. In high-RH experiments, seed particles (sodium chloride and sodium nitrate) were introduced without drying, remaining as liquid particles under the chamber RH."

Line 119: I would recommend changing "high" to long for the description of the lifetime. A long lifetime reads better than high lifetime.

We agree with the reviewer and have changed to "long".

Line 125: How was the H2O2 concentration added calculated? I assume the 30% H2O2 is in water and that would add water vapor to the chamber, was this dried before or is the mass of water added insignificant on the scale of the chamber?

The $H_2O_2$ concentration in the chamber was first estimated based on the known amount of $H_2O_2$ water solution (30%) added into the chamber via a micro-syringe. Then the actual $H_2O_2$ concentration used in the box model was derived by tuning the estimated concentration to match the observed decay rate of NO (in experiment 2b). The amount of added water vapor from the $H_2O_2$ injection was insignificant to affect the chamber RH.

In the revised manuscript, in Line 135, we have changed the text to:

"For low-NO (long $\tau_{bi}$) experiments, ppm levels of $H_2O_2$ were added as the OH precursor, by vaporizing a known amount of 30% $H_2O_2$ solution injected by a micro-syringe. The $H_2O_2$ concentration was derived based on the known photon flux in the chamber and the observed decay rate of NO."

Line 138: Only Sulfate and MSA were permitted to partition to the particle phase. Could the exclusion of DMSO, DMSO2, HPMTF, and MSIA known to be lost via heterogeneous processes add to the disconnect between MSA and sulfate yield in the model and experiment discussed later?

Because the model does not have phase partitioning, when we said "all sulfuric acid and MSA formed is assumed to instantaneously partition to the particle phase", we were specially referring

to comparing the model-predicted MSA and sulfuric acid to the measured MSA and sulfate aerosols.

In the revised manuscript, we have changed to the text to (Line 150):

"The uptake or heterogeneous reactions of other water-soluble species (e.g., DMSO, DMSO$_2$, HPMTF, and MSIA) are not considered in this modeling, though as described below such processes may occur."

Line 141: Adding the LOD for the NOx instrument would be helpful as well as adding the LOD for all the detectable species in the SI table would be greatly appreciated.

The LOD for the NOx instrument is 0.4 ppb for 60 s averaging.

In Line 153, we have added the LOD for the NOx instrument:

"In the low-NO experiment (Exp. 2a) in which the sub ppb-level NO concentration was near or below the detection limit (0.4 ppb) of the NO$_x$ analyzer…."

Line 157: The fate of the sulfene and sulfur PAN are not addressed in this manuscript. Were any observations made across the experiments that could be used to constrain other sulfur oxidation channels? I wonder about the potential for MTMP + NO2 to form other PAN species. The thioacid species is not reference later in the text, nor is it shown in Figure 1. Do the authors have any ideas where this sulfur compound could be originating from? Jernigan et al 2022 found that thioacids could form from the OH oxidation of HPMTF, while Chen et al 2021 and others promote a minor channel where the CH3S* radical could yield a thioformaldehyde (CH2S) capable of oxidizing to a thioacid. I would recommend adding a minor discussion on the state of knowledge concerning these molecules.

CH$_2$SO$_2$ and CH$_3$SO$_6$N are minor products observed in our experiments and therefore their time series are "buried" in the figures. We have mentioned these products in the caption of figure 1.

In Line 175, we have added the following text:

"… as well as CH$_2$SO$_2$ (likely a thioacid, which may be formed as an OH oxidation product of HPMTF (Jernigan et al., 2022a)) and CH$_3$SO$_6$N (likely methanesulfonyl peroxynitrate, formed from CH$_3$S(O)$_2$OO + NO$_2$)."

Line 160: The concentration for HO2 was determined utilizing a model. Do you have any species or HO2 specific products within the chamber that could be used to constrain the model? The loss of H2O2 detected by Iodine CIMS or the formation of ROOH from MTMP + HO2?

Our instrument cannot measure HO2 concentration. The ROOH product ($CH_3SCH_2OOH$) has the same chemical formulae as $DMSO_2$ (both as $C_2H_6SO_2$) and therefore could not be separated.

Line 172: Berndt et al JPCL (2019) stated in their SI that they observed HPMTF with ammonia CIMS.

We thank the reviewer for pointing this out. We have deleted this statement.

Line 175: Was only 3% of sulfur found in the aerosol phase for the Low-NO experiments consistent in both the high and low RH experiments or only the low RH? I would assume 65% RH would increase the water content on the walls and the aerosols which would promote soluble molecules to be lost heterogeneously. I would recommend specifying the RH and NOx state at each point in the main text.

The 3.4% (3.1% - 5.4%) sulfur yield in the aerosol phase was calculated for the dry low-NO experiment after about 6 hrs of equivalent OH exposure. In the high-RH low-NO experiment, the overall sulfur closure was worse than that in the dry experiment, suggesting heterogeneously loss to the chamber wall.

In the revised manuscript, we have modified the text in Line 159:

"Figure 2a-b shows the measured product evolution from Experiments 1 and 2a under the dry condition."

Line 185: You discuss the gas-phase mechanisms to MSA, could aqueous processing of MSIA lead to the formation of MSA? MSIA + OH yields SO2 while MSIA + Oxidant in the aqueous phase yields MSA. Could this lead to a disconnect in the MSA to Sulfate yield?

The reviewer is right that MISA + OH can be a source of MSA in the aqueous phase (Hoffmann et al, 2016). The discussion in this section focuses on gas-phase MSA production under dry experiment. Please see our response earlier on gas-phase MISA + OH reaction that produces MSA.

Line 196: Chen et al ACP (2012) discusses the role of MSA yields under high Nox oxidation of DMS. I recommend looking at their previous work to see if there is any comparisons that could be made.

We thank the reviewer for the suggestion. In the high-NO high-RH experiment in Chen et al (2012), the sulfur yield for sulfuric acid are lower than yields in our work. However, it is unclear whether they reported wall loss-corrected aerosol concentration in all of their experiments. Despite the lower yields, the MSA:sulfate in our work is broadly agreement with that in Chen et al (MSA:sulfate ~ 3).

In the revised manuscript, we have included the following text to better connect our work with other studies:

In Line 173:

"The measured MSA:sulfate is in broad agreement with those reported in Chen et al. (2012)."

Line 207 and Table S2: The iodine CIMS should be sensitive to ROOH species (e.g. HPMTF), while I would assume the harsher ionization of the PTR would induce decomposition of the ROOH. In contrast, the PTR should be able to detect DMSO/DMSO2 while the iodine CIMS would be less sensitive. Could the differing ionization mechanism isolate the isobaric compounds?

This is an interesting idea and potentially useful under some circumstances. Unfortunately we did not see meaningful $C_2H_6SO_2$ (nor DMSO) signal in our I-CIMS possibly due to their very low sensitivity in our instrument. In Table S2, we have listed the instruments used in this work and the species measured by each instrument.

Line 223: The calculation of the isomerization rate does not take into account RO2 + RO2 reactions. Do you have evidence that the RO2 concentration or bimolecular rate is insignificant within the chamber. In addition, MTF could form via RO2 + RO2 reactions. I recommend clarifying this point.

Based on the mechanistic model, the $RO_2 + RO_2$ reactions only represent ~1% of the $RO_2$ sink based on the mechanistic modeling and therefore is not included in $RO_2$ $\tau_{bi}$.

In the revised manuscript, in line 255, we have added:

"MCM modelling suggests that $RO_2 + RO_2$ reactions represent ~ 1% of the $RO_2$ sink in the experiments, and therefore the only bimolecular reactions considered are $RO_2 + NO$ and $RO_2 + HO_2$."

Line 247: Patroescu et al JPC (1996) calculates the absorption cross section for MTF with a focus on the aldehyde photolysis. Would the use of this experimental sulfur containing value compared to that of the MCM value change the fraction of HPMTF and MTF lost by photolysis in your chamber?

In our manuscript, we cite Khan et al. 2021 who use MCM's J14 and J41 to represent the different possible pathways for HPMTF photolysis. As a side note, it appears that they have swapped the two rates, using J41 (ROOH + hv) to represent aldehyde photolysis and J14 (aldehyde + hv) to represent ROOH photolysis; however, this is inconsequential when thinking about total HPMTF loss rate. Under our light conditions, J41 (ROOH + hv) = 1.09e-6 and J14 (aldehyde + hv) = 2.21e-6, giving a total photolysis rate for HPMTF of 3.32e-6 s$^{-1}$. Using the experimental cross section for MTF from Patroescu et al. 1996, we obtain a photolysis rate of

8.96e-7 s$^{-1}$. This is consistent with the conclusion that photolysis is a minor loss pathway of HPTMF in our chamber.

In Line 286, we have added:

"Using the experimental cross section for MTF from Patroescu et al. (1996), we obtain a photolysis rate accounting for less than 2% of HPMTF loss in the chamber."

Line 255: I recommend citing Vermeuel et al EST (2019) and Novak et al PNAS (2021) as they also made this argument using field measurements.

We have included these two citations.

Line 256: I recommend adding a citation here as the formation of MTF could arise from multiple different channels. Does your model support that the ROOH + OH channel as the dominant channel?

Based on the mechanisms, MTF can also be formed via CH3SCH3OO+RO$_2$; however, the ROOH+OH is the major channel under our experimental conditions.

Here we revised the text to (Line 298):

"MTF is formed predominantly as a second-generation DMS oxidation product from CH3SCH2OOH + OH in low-NO conditions in our experiments."

Line 258: The inclusion of the MTF + OH rate constant and its strong agreement with the previous value provides a nice validation to the method calculating the OH rate constant. Is there any reason one could not use a rate comparison method using DMS + OH and/or MTF + OH to solve the HPMTF rate constant?

The rate compare method through the ln (x/x) would remove the need for sensitivity of the species as well as remove the need to use a model for oxidant concentrations. DMS and MTF OH rate constants were determined using classic flow tube experiments with pure sources of DMS/MTF. For this reason, I would put stock in their calculated value. Would this method greatly change the HPMTF + OH value?

I would recommend adding a figure that shows the fit of the HPMTF + OH and MTF + OH rate. This will help the reader see the calculation.

We actually did use a functionally equivalent method but did not explain it well. We calculated [OH] based on the decay of [DMS] after the addition of NO which shuts off MTF and HPMTF production. The decays of these two species are then fit and the rates can be calculated using the

calculated [OH]. We've amended the manuscript and included an additional figure in the supporting information to make this clear:

In Line 290:

"By calculating [OH] using the decay of DMS after the addition of NO, we fit the decay of HPMTF (Figures 4b and S9) to derive $k_{OH+HPMTF}$ of 2.1 (2.0 – 2.2) × 10$^{-11}$ cm$^3$ molec$^{-1}$ s$^{-1}$."

The following figure is now included in the Supplementary Information:

[Figure]

Line 276: What is the ozone concentration within the chamber? Could the ozone concentration and its partition to the particle lead to oxidation of condensed sulfur on the timescale of the experiment? Also, ozone can promote SO2 formation through the CH3S* + O3 reaction.

The ozone concentration is Experiment 4 started from ~ 0 ppb and gradually increased to ~ 60 ppb at its peak concentration during the experiment. It is possible that condense-phase oxidation happened.

Line 281: Could this be an ozone adduct? [IO3 * SO3]?

Yes, it is possible. In the text, this is included in Line 325.

Line 287: Vermeuel et al 2019 found in their SI that HPMTF has a negative humidity dependence at RH higher than 30%. This is a different instrument and voltages, but the trend in the water dependence should be comparable.

We have added Vermeuel et al. (2019) as the reference here.

Line 291: You state that the HPMTF, DMSO and DMSO2 concentrations are not much different from the High and Low RH. Why would the wall ("surfaces") provide a larger sink for the sulfur

species compared to that of the seed particles? You stated earlier the lifetime to the seeds is orders of magnitude greater. I would recommend clarifying "initial yields" and why only the first 6 hours of the experiment was considered (Figure 4).

Here the initial yields mean the yield of products in the first several hours of the OH exposure (within the timescale shown in Figure 4d). Figure 4 compares the $\Delta$product/$\Delta$DMS for dry vs low experiments. Because the dry experiment only proceeded to 6hrs OH exposure (while the high-RH experiments went to 60h OH exposure), only 6 hrs of data were available to be used to make the comparisons. In the first several hours of OH exposure, the timescale of condensable products condensing to seeds particles are shorter than condensing to the wall. However, as the experiment proceeded, particles gradually lost to the chamber wall and therefore, it is expected that vapor wall loss became more prominent.

In the revised manuscript, in the caption of Figure 5, we have added:

"Changes in product concentrations are plotted against change in DMS concentration over the initial 6 hrs of OH exposure when data from both dry and high-RH experiments were available."

In Line 335, we have changed to text to:

"Over these timescales, the initial yields of DMSO, $C_2H_6SO_2$, and HPMTF are not substantially different in the humid and dry cases."

Line 297: Jernigan et al JPCA (2022) found an increased uptake (10x) of HPMTF to deliquesced NaCl aerosols compared to that of dried NaCl. This provides good support that a major sink within your seeded high RH chamber is aerosol uptake. I would recommend comparing the lifetime of HPMTF to aerosol uptake and OH loss utilizing their value.

Sodium nitrate aerosols were used as the seed aerosols in the high-RH high-NO experiment in which negligible amount of HPMTF was formed in the gas-phase. *If* any HPMTF was formed, and we take the results in Jernigan et al (gamma = $(1.6 \pm 0.6) \times 10^{-3}$ for wetted sodium chloride particles) and the surface area of NaCl in our experiment, the uptake rate coefficient would be ~ $3 \times 10^4 \, s^{-1}$ if HPMTF is formed. This rate is on the same order of magnitude with but faster than the loss rate through reacting with OH (~$10^4 \, s^{-1}$).

Sodium nitrate aerosols were used as the seed aerosols in the high-RH low-NO experiment in which HPMTF was formed, however, the uptake coefficient of sodium nitrate is unknown.

Because there is no HPMTF formed in this experiment, we did not calculate the aerosol uptake rate.

Line 302: I recommend citing Vermeuel et al EST (2019) as they pointed out the importance of clouds in controlling the fate, lifetime and concentration of HPMTF.

We have added Vermeuel et al (2019).

Figure 1: Do you mind labeling the major sulfur species you discuss in the main text? In particular, I would highlight DMSO, DMSO2, MSIA, MSA and HPMTF. The boxed DMS and red HPMTF mechanism is helpful, but finding MSA was not trivial. Jernigan et al GRL (2022) has an extended mechanism with thioacids, if you would like to add a route to the sulfene species you detect.

All measured closed-shell products are shown in bold in Figure 1.

Figure 2: Any comments on the loss of sulfur at the start of the experiment in Figure 2b. Where is the sulfur going? Could the initial unconstrained drop be attributed to the coating the walls of the chamber and setting up an equilibrium with the walls?

The initial unconstrained drops may be due to loss of products to surfaces such as the chamber wall or sampling lines. It is likely that there is an equilibrium between the sampling line and the gas phase. This loss of 1 – 2 ppb S was quite consistent across different experiments and was a relatively small portion of the total sulfur budget at the end of the experiments.

In Line 168, we have added the text below:

"The initial dip in the first 2 hours may be due to loss of products to surfaces such as the chamber wall or sampling lines. It is likely that there is an equilibrium between the sampling line and the gas phase. This drop, of 1 – 2 ppb S, represents a relatively small portion of the total sulfur reacted by the end of the experiment."

Reviewer 2:
Ye et al present new laboratory measurements of the OH-oxidation of DMS at high and low NO and high and low RH. The measurements are used to provide better constraints on the isomerization rate of $CH_3SCH_2O_2$ as well as the bimolecular rate for HPMTF+OH. Both of these rates have been reported previously in the literature, but the uncertainty in the prior measurements is much larger than that reported here. One of the more interesting results of the manuscript is the apparent difference in the MSA-sulfate ratio measured at high NO compared to that expected from the model. The paper is well written and is an important contribution to the literature. I recommend that paper be published and that the authors consider the following comments and suggestions:

**Line 77:** Just confirming that tau(bi) includes reactions of $RO_2$ with $RO_2$ (and $HO_2$) and not just NO.

$\tau_{bi}$ in this work includes $RO_2$+NO and $RO_2$+$HO_2$. The $RO_2$+$RO_2$ pathway only represents ~1% of the $RO_2$ sink in our chamber and is not included in $\tau_{bi}$.

In the revised manuscript, in line 255, we have added:

"MCM modelling suggests that $RO_2$ + $RO_2$ reactions represent ~ 1% of the $RO_2$ sink in the experiments, and therefore the only bimolecular reactions considered are $RO_2$ + NO and $RO_2$ + $HO_2$."

**Section 2:** What is the chamber temperature and how constant is it over an experiment. Or more interestingly, what fraction of DMS oxidized proceeds down the H-abstraction pathway.

The chamber temperature is held at 295 K as stated in Line 88. The variation is less than 1 K. ~ 65% of the DMS + OH reaction proceeds to the abstraction pathway (Barnes et al., 2006). This is mentioned in Section 4 in the Supplementary Information.

Reference:
Barnes, I., Hjorth, J. and Mihalapoulos, N.: Dimethyl sulfide and dimethyl sulfoxide and their oxidation in the atmosphere, Chem. Rev., 106(3), 940–975, doi:10.1021/cr020529+, 2006.

**Line 105:** The authors state that: "The quantification of MSA was determined from the AMS tracer ion CH3SO2 + 105 (see SI); this ion is unique to MSA/methylsulfonate, with negligible contributions from other sulfur-containing species (Hodshire et al., 2019; Huang et al., 2015)." I agree in the context of previous experiments and known other S-containing species, but this work (and other recent work) is highlighting that we don't fully understand DMS oxidation and the variety of S-containing species that are produced under atmospheric conditions. It seems possible that this non-specific ion ($CH_3SO_2^+$) could be from molecules other than MSA as we learn about DMS oxidation. Perhaps there is a way to state this in the manuscript?

The reviewer is correct that $CH_3SO_2^+$ could originate from other molecules, but past research suggests that it primarily originates from MSA. It is technically possible to produce this ion from HPMTF with the loss of CHO, but this seems relatively unlikely. In our data, the reference MSA

spectrum (we took separately) explained the organosulfur peaks quite well with minimal residual, suggesting that MSA is the main product formed.

Since we cannot be completely sure that other unknown species do not contribute to this fragment, in the revised manuscript, we've amended Line 106 to read:

"this ion is believed to be unique to MSA/methylsulfonate…"

**Figure 2:** Sulfur closure over the first 1hr (OH exposure) in the low NO, low RH experiment is not very good. It is exceptional after 1 hr. What is happening in this first hour where you are losing 2ppb of DMS, but there is no indication of any sulfur products being formed? Is this a mixing issue?

We do not think it is a mixing issue because the precursors were allowed to mixed well before lights were turned on. Plus, the lights were evenly distributed around the chamber and therefore the chemistry should be spatially homogenous inside the chamber.

In Line 168, we have added the text below:

"The initial dip in the first 2 hours may be due to loss of products to surfaces such as the chamber wall or sampling lines. It is likely that there is an equilibrium between the sampling line and the gas phase. This drop, of $1 - 2$ ppb S, represents a relatively small portion of the total sulfur reacted by the end of the experiment."

**Line 175:** I would suggest citing Jernigan et al 2022 (JPC-A) where they show that HPMTF uptake to dry aerosol particles is small, consistent with the idea that reactive uptake to seed aerosol is insignificant.

We have added this citation in Line 126 when discussing aerosol composition and reactive uptake.

**Line 190:** The MSA-sulfate part of this story is very interesting. I am curious if among the many experiments you have run if there are sufficient experiment-to-experiment differences in $HO_2$ to test the branching between R4 and R5.

We thank the reviewer for this suggestion. In our experiments, we did not intentionally vary $HO_2$ concentration like what we did for the NO. However, the $HO_2$ concentration in the low-NO experiment (~0.17 ppb in Exp. 2) and the high-NO experiment (~ 0.001 ppb Exp. 1) are different enough to shed light to the effect of $HO_2$ on MSA:sulfate. In the low-NO experiment, the measured MSA:sulfate (~ 1.5) and the modeled MSA:sulfate (~ 0.2) towards the end of the experiment are in a much better agreement than that in the high-NO case in which the measured MSA:sulfate is ~ 2.5 and the modeled MSA/sulfate is only ~ 0.001. This qualitatively suggests that the $HO_2$ concentration is one of the factors in driving the branching between R4 and R5 and thus affecting MSA/sulfate. Other possible factors are also discussed in the text.

**Line 226:** The asymptote of the yield curve of 1.5 is an interesting constraint on the CIMS sensitivity to HPMTF. Is this still consistent with S-closure in the low NO and low RH experiment?

If we apply this 1.5 factor to the HPMTF sensitivity used in the low-NO and low RH experiment, the total sulfur closure changed from 90% (64% - 118%) to 116% (77% - 157%), and thus we still have sulfur closure.

**Line 256:** MTF is also formed from the reaction of $CH_2SCH_2O_2$ with $RO_2$.

The reviewer is right. However in the experimental condition in this study, the major pathway for MTF is ROOH+OH and $CH_2SCH_2O_2$ with $RO_2$ is a minor pathway for MTF.

In the revised manuscript, we have changed the text to:

"MTF is formed predominantly as a second-generation DMS oxidation product from CH3SCH2OOH + OH in low-NO conditions in our experiments."

**Line 264:** Why were the high RH experiments carried out over longer timescales and at higher initial DMS concentrations? Does the higher initial DMS concentration have any impact on tau(bi) through RO2+RO2 reactions?

All experiments were carried out using similar total DMS concentration (See Table 1). In the figures, we are showing the DMS consumed (delta DMS) and therefore, for experiments that ran for a longer time, more DMS was consumed. The $RO_2 + RO_2$ reactions should be a minor loss of RO2 in all experiments. The high-RH experiments were originally planned to run for longer timescale (overnight experiments) to better probe multi-generational products, however, long experiments suffered from significant particle wall causing more products lost to the chamber wall.

In Line 115, we have added:

"Total concentrations of DMS introduced to the chamber were similar among all experiments."

In the caption of Figure 2, we have added:

"Note that y axes denote the changes in concentrations of the precursor and products."

In Line 307, the text now reads:

"These experiments were carried out over longer timescales (higher OH exposures) than the corresponding dry experiments to better probe multi-generational products."

**Line 285:** It would be helpful to assess the contribution of the water dependent HPMTF sensitivity on S-closure here. See Veres et al 2020 PNAS (Supplemental figure S8) for a rough idea of how much of an effect this might have.

The $H_2OI^-$ : $I^-$ for the dry and high-RH experiments were 0.027 and ~0.04, respectively. These are lower than values in Veres et al, probably due to different instrument setup. Here we have added Veres et al as a reference for discussing the decreased HPMTF sensitivity with increasing RH.

**Line 300:** I think Vermeuel et al 2020 first showed the role of clouds/fog on HPMTF with field measurements.

This citation has been added.

**Line 320:** I agree that HPMTF delays the formation of $SO_2$, but it could accelerate the formation of sulfate if HPMTF multiphase chemistry is an efficient pathway to sulfate (as suggested by Novak et al).

In the revised manuscript (Line 368), we have changed the text to:

"In particular, the formation of HPMTF from $RO_2$ isomerization suppresses (or at least delays) the gas-phase formation of $SO_2$, sulfate and MSA."

**Table 1:** It appears that these experiments were run with a large number of different seed particles. There is no discussion of the effect of the seed particle on the sulfur product distribution (and the MSA-sulfate ratio).

In the revised manuscript, we have added the following text in Line 121:

"In these experiments, non-sulfate seeds were used to avoid interferences when quantifying secondary sulfate in the aerosols. For low-RH experiments (experiments 1-3), nitrate seed particles were used, since dry nitrate particles are expected to be chemically inert. For the high-RH high NO experiment (Exp. 4), NaCl particles were used. As discussed below, major products are similar to those in the high-NO dry experiment, suggesting that the NaCl seed particles in Exp. 4 have little to no effect on the product distribution in these experiments. More studies are needed to constrain the effects of different seed particle on the reactive uptake of DMS oxidation products (Jernigan et al., 2022b)."

Reviewer 3:

Ms.No. acp-2022-566

The authors describe experimental findings from the OH-initiated oxidation of DMS conducted in a chamber under almost atmospheric conditions. Bimolecular RO2 lifetime was varied in a wide range from > 10 s to < 0.1 s by changing NO in the reaction gas from 10 ppt up to 50 ppb. A suite of analytical techniques was used for comprehensive analysis of the resulting product distribution in the gas phase as well as of the formed aerosol products on a seed. Reaction conditions for the long bimolecular lifetime, standing for the pristine atmosphere, allowed studying the product formation in the abstraction channel governed by CH3SCH2O2 isomerization. The rate coefficient of this isomerization step was re-investigated being in good agreement with other experimental values. Especially the fantastic sulfur closure in the measured products is worth mentioning, which demonstrates the accuracy in conducting the experiment and in product analysis.

This work represents a next, very nice piece of work from this group on the understanding of chemical processes in the reaction of OH with DMS, here especially for the 1st generation gas-phase products.

The manuscript is well written and suitable for publication in this journal. Here only a few minor comments:

We thank the reviewer for the positive comments and the suggestions. We have addressed them in the responses below and edited the manuscript accordingly.

1) While the agreement between experiment and model in the low-NO case is very good, there are clear differences in the high-NO case. Here, the MSA production seems to be strongly underestimated by the mechanism. High-NO experiments have been already done 20 years before by the Wuppertal group and by others, mostly with relatively high reactant concentrations. Are the findings from the present work consistent with the former results? Can we learn anything from the comparison?

In the revised manuscript, we have included more discussions to connect to previous studies. Specifically, we extended the discussions on MSA production mechanisms reported in previous studies and included those mechanisms in the box model for model-measurement comparisons.

In Line 223, we have added the following paragraph:

Another potential source of MSA is the OH-initiated oxidation of MSIA by OH (Yin et al., 1990; Lucas and Prinn, 2002; von Glasow and Crutzen, 2004; Wollesen de Jonge et al., 2021; Shen et al., 2022). This pathway is currently not included in the MCM, which has MSIA reacting with OH to form $SO_2$ and $CH_3$ (Figure 1). It has been suggested (Yin et al., 1990) that the reaction may occur via abstraction of the acidic hydrogen:
$CH_3S(O)OH$ (MSIA) + OH → $CH_3S(O)O$ + $H_2O$         (6)
As shown in Figure 1, the resulting $CH_3S(O)O$ radical may react with ozone to form $CH_3S(O)_2O$, which can react further to form MSA or $SO_3$ (reactions 4-5). However, inclusion of this reaction in the model increases MSA formation only slightly, and the model-measurement discrepancy

remains large (Figure S5). Alternatively, OH might add to MSIA (Lucas and Prinn 2002; Arsene et al., 2002), forming the intermediate $CH_3SO(OH)_2$ that can react with $O_2$ to produce MSA:

$$CH_3S(O)OH \text{ (MSIA)} + OH \xrightarrow{M} CH_3S(O)(OH)_2 \tag{7}$$
$$CH_3S(O)(OH)_2 + O_2 \rightarrow CH_3S(O)_2(OH) \text{ (MSA)} + HO_2 \tag{8}$$

Including these reactions into the mechanism, using the rate constant for MSIA + OH suggested by the MCM ($9 \times 10^{-11}$ $cm^3$ $molec^{-1}$ $s^{-1}$) substantially increases the predicted MSA, but at the same time decreases the predicted $SO_2$ concentration, worsening the model-measurement agreement for $SO_2$, and does not change predicted sulfate formation, leading to an overestimate in total aerosol production (Figure S5). Taken together, while the OH oxidation of MSIA (reactions 6-8) may contribute to MSA formation, it is not the only (or major) source for the MSA model-measurement discrepancy in the present experiments.

References:
Yin, Fangdon, et al. "Photooxidation of dimethyl sulfide and dimethyl disulfide. II: Mechanism evaluation." *Journal of Atmospheric chemistry* 11(4)365-399, 1990.

Von Glasow, R., and P. J. Crutzen. "Model study of multiphase DMS oxidation with a focus on halogens." *Atmospheric Chemistry and Physics* 4(3) 589-608, 2004

Wollesen de Jonge, Robin, et al. "Secondary aerosol formation from dimethyl sulfide–improved mechanistic understanding based on smog chamber experiments and modelling." *Atmospheric Chemistry and Physics* 21(13) 9955-9976, 2021

Shen, Jiali, et al. "High Gas-Phase Methanesulfonic Acid Production in the OH-Initiated Oxidation of Dimethyl Sulfide at Low Temperatures." *Environmental science & technology* (2022).

Lucas, D. D. and Prinn, R. G.: Mechanistic studies of dimethylsulfide oxidation products using an observationally constrained model. Journal of Geophysical Research: Atmospheres, 107(D14), ACH-12, 2002.

Arsene, C., Barnes, I., Becker, K. H., Schneider, W. F., Wallington, T. T., Mihalopoulos, N. and Patroescu-Klotz, I. V.: Formation of methane sulfinic acid in the gas-phase OH-radical initiated oxidation of dimethyl sulfoxide, Environ. Sci. Technol, 36(23), 5155–5163, 2002.

2) Product distributions are presented for very long and very low bimolecular RO2 lifetimes, > 10 s and < 0.1 s. But what happens in between? The RO2 lifetime was varied in order to estimate the rate coefficient of CH3SCH2O2 isomerization based on HPMTF yields. Could the authors provide more information on the other products as a function of lifetime?

It is challenging to maintain a relatively steady intermediate level of NO (thus intermediate $RO_2$ lifetime) in the chamber because the NO is continuously consumed by $RO_2 + NO$. To study

intermediate NO level chemistry, it requires continuous injection of a small amount of NO into the chamber, which is unfortunately not how these experiments were run.

However, Experiment 3 can be viewed as an "in-between" case in which we varied $RO_2$ lifetime several times within the course of the experiment by pulse injections of NO. The product distribution is presented in Figure S8. A reduced yield of HPMTF and an increased yield of aerosols (MSA and sulfate) compared to the low-NO experiment were observed, in qualitative agreement with the trend observed in the high-NO and low-NO experiments.

3) k(CH3SCH2O2 isomerization) was obtained in an indirect way relative to k(RO2 + NO), right? What was the used value of k(RO2 + NO)? It´s important to have a comparison with other studies. Was the CH3SCH2O2 + HO2 reaction neglected in the analysis? Please explain more in detail the determination of k(isomerization).

The detailed determination of $k_{isom}$ is given in Section 4 in the Supplementary Information. CH3SCH2O2 + HO2 reaction was not neglected. $k(CH_3SCH_2O_2 + NO)$ and $k(CH_3SCH_2O_2 + HO_2)$ were summed to estimate the bimolecular rate of the $CH_3SCH_2O_2$ radical to obtain $k(CH_3SCH_2O_2$ isomerization):
$k(CH_3SCH_2O_2 + NO) = 4.9 \times 10^{-12} * \exp(260/T)$;
and
$k(CH_3SCH_2O_2 + HO_2) = 2.91 \times 10^{-13} * \exp(1300/T) * 0.387$

These values were directly taken from MCMv3.3.1. The $k(CH_3SCH_2O_2 + NO)$ is also recommended in JPL Chemical Kinetics. At 295 K (this experiment), $k(CH_3SCH_2O_2 + NO)$ used in this study is $1.18 \times 10^{-11}$ cm$^3$ molec$^{-1}$ s$^{-1}$, in good agreement with $8.0 \pm 3.1 \times 10^{-12}$ cm$^3$ molec$^{-1}$ s$^{-1}$ reported in Urbanski et al (1997) and $1.20 \times 10^{-11}$ cm$^3$ molec$^{-1}$ reported in Turnipseed et al (1996).

Reference:
1. Turnipseed, A. A.; Barone, S. B.; Ravishankara, A. R. Reaction of OH with dimethyl sulfide. Products and mechanisms. J. Phys. Chem. 1996, 100, 14703-14713.

2. Urbanski, S. P.; Stickel, R. E.; Zhao, Z.; Wine, P. H. Mechanistic and kinetic study of formaldehyde production in the atmospheric oxidation of dimethyl sulfide. J. Chem. Soc. Faraday Trans. 1997, 93, 2813-2819.

3. Burkholder, J. B., et al. *Chemical kinetics and photochemical data for use in atmospheric studies: evaluation number 18*. Pasadena, CA: Jet Propulsion Laboratory, National Aeronautics and Space Administration, 2015, 2015.

Moreover, CH3SCH2O2 isomerization leads to the next RO2 radical HOOCH2SCH2O2 after O2 addition. And the next isomerization step in HOOCH2SCH2O2 finally ends up

in HPMTF + OH. What is known about the rate of HOOCH2SCH2O2 isomerization? Is HOOCH2SCH2O2 + NO distinctly slower even for the high NO addition of 50 ppb? Otherwise it must be considered in the data analysis because HPMTF was taken for the determination of k(CH3SCH2O2 isomerization), not the direct isomerization product HOOCH2SCH2O2.

The second H transfer reaction is estimated to be much faster than the first one (Wu et al., 2015; Crounse et al., 2013) and therefore the first H transfer is the rate-limiting step. Under high NO condition, the first H transfer reaction cannot compete with the bimolecular reaction with NO, and essentially no HPMTF forms.

In the revised manuscript, in Line 57, we have added:

"The alkyl radical derived from Reaction 3 will react with $O_2$ to form $OOCH_2SCH_2OOH$, which will undergo a second isomerization reaction at a rate substantially faster than that of Reaction 3 (Wu et al., 2015; Crounse et al., 2013), forming hydroperoxymethyl thioformate (HPMTF, $HOOCH_2SCHO$), as shown in Figure 1."

4) It is not the first time that HPMTF was detected by ammonium CIMS. In Berndt et. al a suite of ionization schemes was used for HPMTF monitoring including ammonium (with the PTR3), see fig. S3 and explanation in the main body.

In the revised manuscript, we have removed this sentence.

Community comments:

Dear authors,

I have read your paper with great interest, and I have a comment particularly regarding your statement in the introduction saying that "very few studies of the entire multiphase and multistep reaction system have been conducted...".

I would like to make you aware of three papers we have recently published/submitted on DMS oxidation and related aerosol formation in our group (Rosati et al., 2021; Rosati et al., 2022; Wollesen de Jonge et al., 2021).

Our studies focused on the pure new particle formation from the DMS+OH reaction at low NOx, high and low relative humidity, different DMS concentrations and different temperatures. A particular focus was put on the measurement of MSA by HR-ToF-MS in our experiments. As described in detail in Wollesen de Jonge et al. (2021) we also employed a model that implemented new reactions in the MCMv3.3.1 and the formation of HPTMF.

As you use different seed aerosols I was wondering about a few points:

- How many and what mass of seed did you use during the experiments?
- Did all oxidation products condense on the pre-existing seed aerosols or did you simultaneously observe new particle formation?
- Did the use of the different seeds (i.e. ammonium nitrate, sodium nitrate, sodium chloride) affect the results?

It would be interesting to see a comparison/discussion of your results with our chamber experiments as far as possible given the different conditions.

We thank Dr. Rosati for the comments. The seed particle concentrations were on the order of 10's $\mu$g m$^{-3}$ when oxidation started. New-particle formations were observed in most of the experiments. As we stated in the manuscript, condensation timescales (seconds to 10's of seconds) onto the seed aerosols were much shorter than the condensation timescale of low-volatility species onto the chamber wall ($\sim$ 2000 s). With new-particle formation occurring, there were even more aerosol surface area available for vapor condensation, except for Experiment 5, in which vapors wall loss was significant due to particle wall loss over the longer experimental timescale.

Our paper primarily focuses on the gas-phase chemistry of DMS OH oxidation and investigates the product distribution under different peroxy radical regimes. In the revised manuscript, we have included Wollesen de Jonge et al. (2021) as a reference for the discussion of the importance of MISA + OH as a pathway in MSA formation.

In Line 205, we have added:

"This suggests the mechanism may underestimate the rate of MSA formation (a result consistent with recent studies (Wolleson de Jonge et al. 2021; Shen et al. 2022), and/or overestimate the rate of sulfuric acid formation."

In Line 223:

"Another potential source of MSA is the OH-initiated oxidation of MSIA by OH (Yin et al., 1990; Lucas and Prinn, 2002; von Glasow and Crutzen, 2004; Wollesen de Jonge et al., 2021; Shen et al., 2022). This pathway is currently not included in the MCM, which has MSIA reacting with OH to form $SO_2$ and $CH_3$ (Figure 1)."

In terms of seed particles, in the revised manuscript, we have added the following text in Line 121:

"In these experiments, non-sulfate seeds were used to avoid interferences when quantifying secondary sulfate in the aerosols. For low-RH experiments (Exp. 1-3), ammonium nitrate seed particles were used, since dry ammonium nitrate particles are expected to be chemically inert. For the high-RH high NO experiment (Exp. 4), NaCl particles were used. As discussed below, major products are similar to those in the high-NO dry experiment, suggesting that the NaCl seed particles in Exp. 4 have little to no effect on the product distribution in these experiments. More studies are needed to constrain the effects of different seed particle on the reactive uptake of DMS oxidation products (Jernigan et al., 2022b)."